# Enhancer hijacking at the *ARHGAP36* locus is associated with connective tissue to bone transformation

Uirá Souto Melo [1,2] ✉, Jerome Jatzlau[3], Cesar A. Prada-Medina[1], Elisabetta Flex [4], Sunhild Hartmann[1], Salaheddine Ali [1], Robert Schöpflin[1], Laura Bernardini[5], Andrea Ciolfi [6], M-Hossein Moeinzadeh[7], Marius-Konstantin Klever[1,2], Aybuge Altay [7], Pedro Vallecillo-García[3], Giovanna Carpentieri[6], Massimo Delledonne [8], Melanie-Jasmin Ort[3,9,10], Marko Schwestka[11,12], Giovanni Battista Ferrero[13], Marco Tartaglia [6], Alfredo Brusco [14,15], Manfred Gossen[11,12], Dirk Strunk [16], Sven Geißler [9,10,12], Stefan Mundlos [1,2], Sigmar Stricker [3], Petra Knaus [3], Elisa Giorgio[17,18] ✉ & Malte Spielmann[1,19,20] ✉

Heterotopic ossification is a disorder caused by abnormal mineralization of soft tissues in which signaling pathways such as BMP, TGFβ and WNT are known key players in driving ectopic bone formation. Identifying novel genes and pathways related to the mineralization process are important steps for future gene therapy in bone disorders. In this study, we detect an inter-chromosomal insertional duplication in a female proband disrupting a topologically associating domain and causing an ultra-rare progressive form of heterotopic ossification. This structural variant lead to enhancer hijacking and misexpression of *ARHGAP36* in fibroblasts, validated here by orthogonal in vitro studies. In addition, *ARHGAP36* overexpression inhibits TGFβ, and activates hedgehog signaling and genes/proteins related to extracellular matrix production. Our work on the genetic cause of this heterotopic ossification case has revealed that *ARHGAP36* plays a role in bone formation and metabolism, outlining first details of this gene contributing to bone-formation and -disease.

Bone development and growth is a continuous process that starts during prenatal development and plays an important role throughout life in regenerating and repairing the skeleton. In some rare cases, soft tissues can mineralize due to anomalies in the repair mechanism resulting in heterotopic ossification (HO), which can be caused by genetic and non-genetic factors[1]. Non-genetic induced HO often shows ectopic bone formation as a consequence of soft tissue trauma after injury, resulting in the ossification of connective tissue[1]. Genetic HOs are often caused without trauma/injury, and very few genes have been associated with this condition. The most studied genetic form of HO is the autosomal

dominant fibrodysplasia ossificans progressiva (FOP)[2], with a prevalence of one in 2.5 million births[3]. In FOP patients, extra-skeletal bone formation (either spontaneous or in response to trauma) begins during early childhood and progresses throughout life[4,5]. Aside from the progressive ossification of muscle, tendon, and ligaments, the classical FOP phenotype presents feet and spine deformity and/or hearing loss[6,7].

Gain-of-function mutations in the activin receptor type 1 (*ACVR1*), a type I bone morphogenic protein (BMP) receptor, cause FOP[2]. BMP and transforming growth factor beta (TGFβ) signaling pathways are key players in natural bone formation and also contribute to ectopic

osteo-differentiation in HO patients[8]. Pathogenic variants in genes related to bone-development pathways are associated with other forms of HO; for instance, *GNAS1* loss- or gain-of-function mutations account for the protein kinase A (PKA) and WNT pathways activation, respectively[9,10]. TGFβ, BMP, and WNT pathways converge on RUNX2, a transcription factor associated with osteoblast differentiation, considering that their crosstalk can either promote or attenuate osteoblast maturation[11]. Therefore, a convergent mechanism for bone-related pathways has been proposed to explain the majority of HO forms[12].

In this study, we report on an isolated, ultra-rare, rapidly progressive form of HO leading to premature death due to extensive ossification of connective tissue. We showed that this unique phenotype is caused by an inter-chromosomal insertional duplication, disrupting a topologically associating domain (TAD) structure of the X chromosome, resulting in enhancer hijacking and *ARHGAP36* misexpression. Our work on the genetic cause of this disease revealed that *ARHGAP36* activation interferes with gene expression of important pathways related to bone formation and heterotopic ossification.

## Results

### Clinical findings and disease progression of a rare case of heterotopic ossification (HO)

We reported a female individual of a non-consanguineous family presenting with an isolated, ultra-rare, rapidly progressive form of HO (Fig. 1a). At 5 months old; the parents noticed decreased body mobility, especially of distal joints, including hands, feet, wrists, and ankles, which could be attributed to marked calcifications in the surrounding tissues. At 8 months, magnetic resonance imaging (MRI) scans showed calcifications progression with almost complete apparent ossification of the posterior longitudinal ligament and involvement of masticatory and inter-costal muscles[13]. At 3 years, a CT scan revealed further progression of the disease with spots of calcium deposition in muscles of the jaw, the limbs, over the shoulders and scapulae, and especially in the hips and pelvis. After 5 years, most of her skeletal muscles were turned into bone, inevitably causing her death at the age of 8 due to respiratory complications (Fig. 1a).

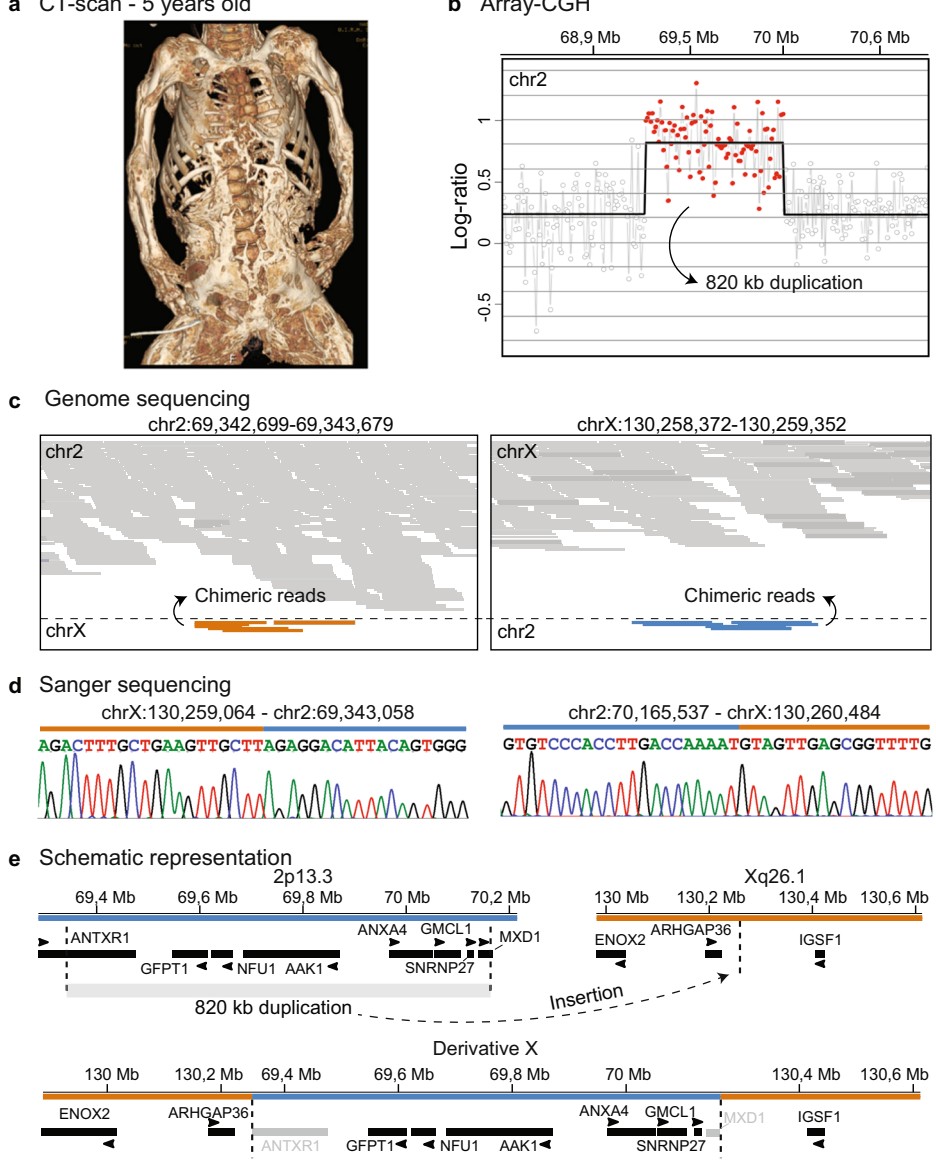

**Fig. 1 | Extreme case of heterotopic ossification. a** Computed tomography (CT) scan at the age of 5 years shows the muscle-to-bone transformation in the proband. **b** Array-CGH detected an 820 kb duplication on chr2. **c** Whole genome sequencing mapped the duplication to the chrX (orange: chrX; blue: chr2). **d** Sanger sequencing mapped the breakpoints at the base pair level. **e** Schematic representation of the duplicated genes from chr2 and the insertion at the *ARHGAP36/IGSF1* locus in chrX.

### *De novo* microduplication on chromosome 2 inserted into Xq26

While several genes have been associated with HO/ossification disorders (Table S1)[2,14–21], trio whole exome sequencing did not reveal any pathogenic variant, suggesting a yet unknown underlying mechanism. No deleterious variants in functionally relevant genes based on the clinical presentation, however, were identified. Next, array-CGH detected a heterozygous ~820 kb duplication on 2p13.3 encompassing eight coding genes: two partially (*ANTXR1* and *MXD1*) and six completely open reading frame (ORF)-duplicated (*GFPT1*, *NFU1*, *AAK1*, *ANXA4*, *GMCL1*, and *SNRNP27*) (Fig. 1b). Fluorescence in situ hybridization (FISH) revealed *de novo* inheritance and insertion of the duplicated fragment on Xq26.1 [der(X)ins(X;2)(q26.1;p13.3)] (Fig. S1a, b). Trio whole genome sequencing confirmed the previous observations, and Sanger sequencing further mapped the breakpoint at the insertion site at base pair resolution (Fig. 1c–e; Fig. S1c).

### Restructured 3D genome architecture on derivative X creates a novel chromatin domain granting enhancer hijacking

Copy number variation of genomic regions (e.g., deletions and duplications) can influence gene dosage per se, while copy neutral variants (e.g., cut and paste insertions) can disrupt genes and have additionally the potential to disrupt or create new enhancer–promoter interactions[22]. Structural variants spanning TAD boundaries are at risk of granting enhancer adoption, therefore causing gene dysregulation or misexpression[23,24]. To evaluate the local chromatin landscape on the derivative chrX (der(X)), we performed chromosome conformation capture analysis (Hi–C) in proband-derived fibroblasts and observed ectopic signals on the Hi-C trans-map (chr2–chrX) (Fig. 2a; Fig. S2a). The 820 kb duplication was inserted into a ~0.8 Mb TAD containing two coding genes, *ARHGAP36* and *IGSF1*, resulting in loss of chromatin contacts in the proband Hi–C (Fig. 2b). We next created a customized der(X) containing the "chrX–chr2–chrX" genomic sequence and re-mapped the genome-wide Hi–C reads, in both proband and control (Fig. 2c, Fig. S2a).

Based on visual inspection of the custom der(X) cis-map, two possible novel chromatin domains could be formed: TAD #1 on the left breakpoint by connecting *ARHGAP36* to partially ORF-less *ANTXR1* region; and TAD #2, a domain containing *ANXA4*, *GMCL1*, *SNRNP27*, ORF-less *MXD1*, and *IGSF1* (Fig. 2c). Control custom-map shows blank spaces in TADs #1 and #2, as expected because there is no physical connection between chr2 and chrX in this sample. On the other hand, the proband custom-map shows a novel chromatin domain (Shuffled-TAD) on der(X) left breakpoint (TAD #1) (Fig. 2c, d). The Shuffled-TAD contains *ARHGAP36* and putative enhancers on the *ANTXR1* gene body, while the right breakpoint (TAD #2) shows weak novel chromatin interaction. We next evaluated the gene expression of all candidate genes within the duplication and surrounding the insertion on chrX.

### The chr2 duplicated region partially maintains its chromatin activity despite being inserted in an inactive part of the genome

First, we excluded the ORF-duplicated genes as candidates to be causative for the phenotype (see the comprehensive explanation of the rationale behind discarding these genes as candidates in Supplementary Information). Second, in order to assess the chromatin activity in both chr2 and chrX loci, we took advantage of an in-house ChIP-seq dataset of H3K4me1, H3K4me3, and H3K27ac epigenetic marks generated from two different MSC types and fibroblasts from healthy donors (Fig. S2b). Visual inspection of local chr2 and chrX regions revealed two regimes of chromatin signature shared by wild-type MSCs and fibroblasts: the chr2 locus contains several active epigenetic marks, named "active chromatin domain," whereas the *ARHGAP36* and *IGSF1* TAD on chrX showed no epigenetic signal (inactive chromatin domain) (Fig. S2c). Lastly, we performed fibroblast RNA-seq expression data as a readout of the chromatin status at the der(X) region. The ORF-duplicated *GFPT1*, *NFU1*, and *AAK1* genes were upregulated in the proband, indicating that the chr2 active chromatin region kept its activity on der(X)

(Fig. 3a). The remaining ORF-duplicated genes (*ANXA4*, *GMCL1*, and *SNRNP27*) as well as *IGFS1*, all located in TAD #2, show no difference in expression (Fig. 3a, b). We further excluded skewed X chromosome inactivation as an explanation for the silencing of duplicated genes on TAD #2 (see Supplementary Information; Fig. S3).

While the gene expression profiles in the chromatin domain #2 did not reveal any changes, *ARHGAP36* - which is located in the Shuffled-TAD #1, was upregulated in RNA of the proband fibroblasts; additionally, western blot confirmed activation of ARHGAP36 protein in the proband samples (Fig. 3b, c). This upregulation of *ARHGAP36* suggests an enhancer hijacking of regulatory elements located within the inserted *ANTXR1* gene body. Interestingly, *ARHGAP36* shows the highest upregulation in the whole RNA-seq dataset (Fig. S4a). We next ran the CRUP tool to infer regulatory elements from epigenetic marks to identify potential MSCs and fibroblast enhancers in both loci. CRUP analysis revealed several putative enhancers located within the Shuffled-TAD #1 (Fig. 3a).

### Role of ARHGAP36 in bone formation

*ARHGAP36* encodes a RhoGAP signal transduction protein[25] that activates the non-canonical hedgehog (HH) pathway[26] and works as a potent antagonist of the protein kinase A (PKA) signaling (Eccles et al., 2016). Interestingly, *Arhgap36* overexpression in mouse fibroblast-like cells induced osteoblasts differentiation via the HH pathway[26]. Therefore, we investigated the role of *ARHGAP36* by analyzing the differentially expressed genes (DEGs) in proband fibroblast RNA-seq data. First, expression data analysis showed 3.1% of DEGs in the proband compared to the controls (Fig. S4c). Next, we observed an enrichment of DEGs for HH (20.5%; chi-square $p < 0.00001$), BMP-TGFβ (10.8%; $p = 0.000081$), and WNT (15.9%; $p < 0.00001$) related genes (Fig. S4b, c), important pathways associated with different forms of HO[8]. The pathway enrichment analysis detected in the proband might be related to *ARHGAP36* activation or to the effect of background genetic variation, independent of *ARHGAP36* overexpression.

Therefore, we transiently transfected *ARHGAP36* (and GFP as control) in human MSCs to evaluate the global expression profile in response to *ARHGAP36* overexpression (Fig. 4a). DEG analysis showed a high variance between ARHGAP36 and GFP transfected cells, where ten clusters of co-expressed genes (hereafter named K1–K10) were observed (Fig. S5). We selected four co-expression clusters showing high variability in ARHGAP36-transfect samples and minor variance among GFP ones (K1, K3, K4, and K5) for further analysis (Fig. 4b). Interestingly, gene ontology analysis revealed that K1 and K3 contain a set of downregulated co-expressed genes that are part of the BMP-TGFβ pathways (K1) and are involved in osteogenesis regulation (K3) (Fig. 4c). Many TGFβ pathway target genes are downregulated in K3, such as *CEBPD*, *DAPK1*, *GBP2*, and *SELENOP* (Fig. S6). Cluster K4 shows enrichment for tight junction (cell-adhesion) genes, all characteristic of extracellular matrix (ECM) production.

### *ARHGAP36* overexpression inhibits TGFβ and BMP pathway activities

In order to validate the results showing reduced BMP-TGFβ pathway activity upon *ARHGAP36* induction, we transfected NIH/3T3 (fibroblasts) and C2C12 (myoblast-like) with empty vector (control), human *ARHGAP36* (h*ARHGAP36*) or mouse *Igsf1* (m*Igsf1*; positive control) for dual-luciferase reporter gene assay (Fig. 5a; Fig. S7). To measure the activity of the BMP-TGFβ1 pathway, we co-transfected either a BMP- or a TGFβ-sensitive reporter (BRE2-Luc or CAGA12-Luc, respectively), stimulated with either BMP2 or TGFβ1 and measured luciferase activity (Fig. 5a; Fig. S7). Cells transfected with h*ARHGAP36* showed significant decreased TGFβ pathway activity ($p < 0.05$) in both cell lines and concentrations, more prominently at higher TGFβ1 concentration in C2C12 (Fig. 5a). m*Igsf1* overexpression, here used as a positive control, inhibits the TGFβ pathway only in myoblast-like cells at 0.1 ($p < 0.05$)

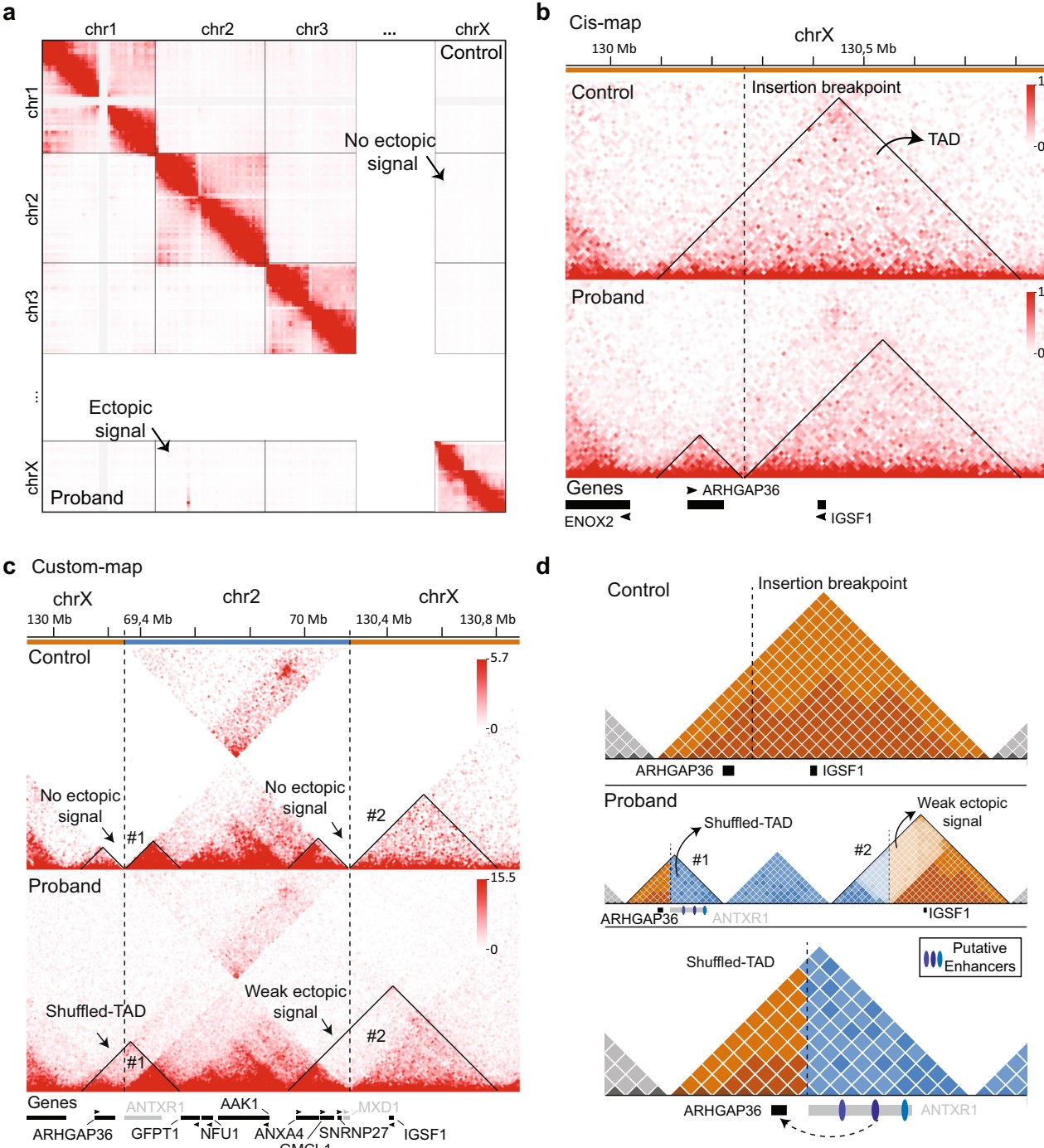

**Fig. 2 | Formation of a novel chromatin domain in the proband Hi-C. a** Hi-C map showing ectopic signal in the "chr2–chrX" trans-map of the proband. **b** Hi-C cis-map shows the topologically associating domain (TAD) containing *ARHGAP36* and *IGSF1*. In the proband, the insertion breakpoint (vertical dashed line) reduces the chromatin interaction within this chromatin domain. **c** Customized "chrX–chr2–chrX" map in control shows blank spaces between chr2–chrX contacts. Two putative novel chromatin domains left (#1) and right breakpoints (#2), contain

no mapped Hi-C reads in control, as expected. On the other hand, proband custom-map shows Hi-C reads in #1, indicating not only physical proximity between chr2–chrX but also the formation of a new chromatin domain (Shuffled-TAD). A weak Hi-C signal is observed at the right breakpoint (#2). **d** Schematic representation of the rearranged der(X). Putative enhancers located on the *ANTXR1* gene body may ectopically activate *ARHGAP36* in a cell-type specific manner.

and 0.2 nM concentrations ($p < 0.01$). Interestingly, the BMP-reporter activity is significantly influenced by the transient expression of either h*ARHGAP36* or m*Igsf1* in both cell lines at the higher BMP2 concentration (Fig. S7).

In order to evaluate the BMP/TGFβ pathway activity in proband and control fibroblasts, we stimulated with TGFβ1 (0.2 nM), BMP2 (5 nM), or Activin A (3 nM) for three-time points, and measured for

SMAD1/3/5 phosphorylation levels via western blot. We observed slightly reduced TGFβ1-induced phospho-SMAD3 levels (i.e., pSMAD3/GADPH ratio) for proband cells ($p = 0.05$) after 30 min of exposure, when compared to three independent controls (Fig. 5b; Fig. S8c), and a significant reduction after 60 min ($p < 0.05$) (Fig. 5c). We excluded that the reduction of phosphorylated SMAD3 was caused by low levels of SMAD3 in the proband, as we did not observe

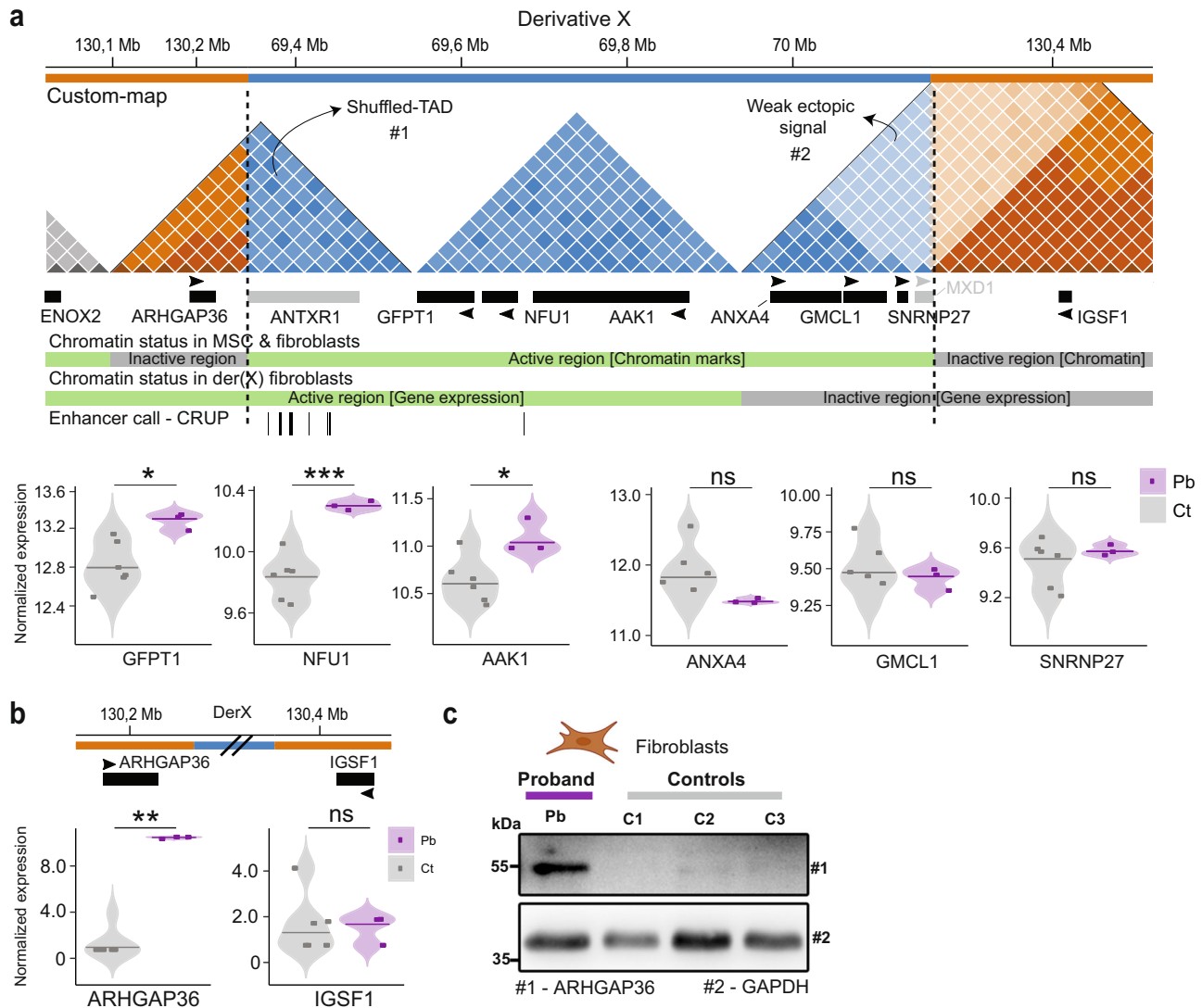

**Fig. 3 | Chromatin activity and gene expression evaluation of der(X).**
**a** Schematic representation of the customized "chrX–chr2–chrX" 3D genomic landscape (top). Active and inactive chromatin domains were classified based on histone mark signals from MSCs (middle) (Fig. S2c). MSC enhancers were called by the enhancer tool CRUP and showed enrichment at the *ANTXR1* gene body. RNA-seq in fibroblasts in the proband and controls (bottom). Purple color represents the proband sample, and gray represents the controls. Expression data is VST normalized. Pb: proband; Ct: control. **b** *ARHGAP36* is upregulated (activated) in the proband in comparison to controls, while *IGSF1* is not expressed in all tested fibroblasts. Statistical significance within the indicated groups was calculated using the Wald test (DESeq2) and Benjamini-Hochberg multiple comparisons test with a 95% confidence interval of the fitted general linear model; *p*-value *<0.05; **<0.01; ***<0.001. **c** Western blot confirmed activation of ARHGAP36 in the proband fibroblasts (*n* = 2 technical replicates). Source data are provided as a Source Data file.

differences in total SMAD3 protein levels in comparison to controls (Fig. S8d). Moreover, we performed a western blot to evaluate the non-canonical TGFβ signaling by measuring the phosphorylation levels of TAK1, in which we observed no changes in proband cells compared to controls (Fig. S8e). However, we evaluated the phosphorylation of the TAK1-downstream target p38 (pp38) and observed a significant increase of p38 phosphorylation in proband cells compared to controls, independent of TGFβ1-induced TAK1 phosphorylation (Fig. S8e). BMP2-induced pSMAD1/5 showed no difference between patient and control samples (Fig. 5c; Fig. S8b, c). In contrast to FOP, where Activin A induces SMAD1/5 phosphorylation, proband fibroblasts here treated with Activin A showed neither differences in SMAD1/5 nor in SMAD3 phosphorylation (Fig. S8b). Taken together, the TGFβ signaling pathway, which is important for chondrogenesis and osteogenesis, is marginally reduced in proband fibroblasts upon TGFβ1 stimulation and in transiently *ARHGAP36* overexpressing cells.

**Human MSCs and chondrocytes overexpressing *ARHGAP36* show increased ECM deposition**
HO-related diseases are characterized by the capability of connective tissues to turn into bone. Here we transiently transfected human connective tissue cells (MSCs and chondrocytes) from healthy donors with *ARHGAP36* (and *GFP* as control) to evaluate this gene function during osteogenic differentiation (Fig. S9a). On day 1, osteogenic markers (COL1A1, COLX, and RUNX2) are upregulated in *ARHGAP36* transfected cells when compared to the control (Fig. S9b–e). COL1A1, one of the most abundant proteins in bone, shows strong expression after day 1 of differentiation in both MSCs and chondrocytes; COLX appears to be upregulated only in chondrocytes transfected with *ARHGAP36* in comparison to control (Fig. S9b). The same pattern is observed up to day 4 of differentiation in both cell types for COL1A1 and RUNX2 (Fig. S9d, e). Collectively, *ARHGAP36* expression in MSCs and chondrocytes induces early osteogenic differentiation markers and extra-cellular matrix proteins.

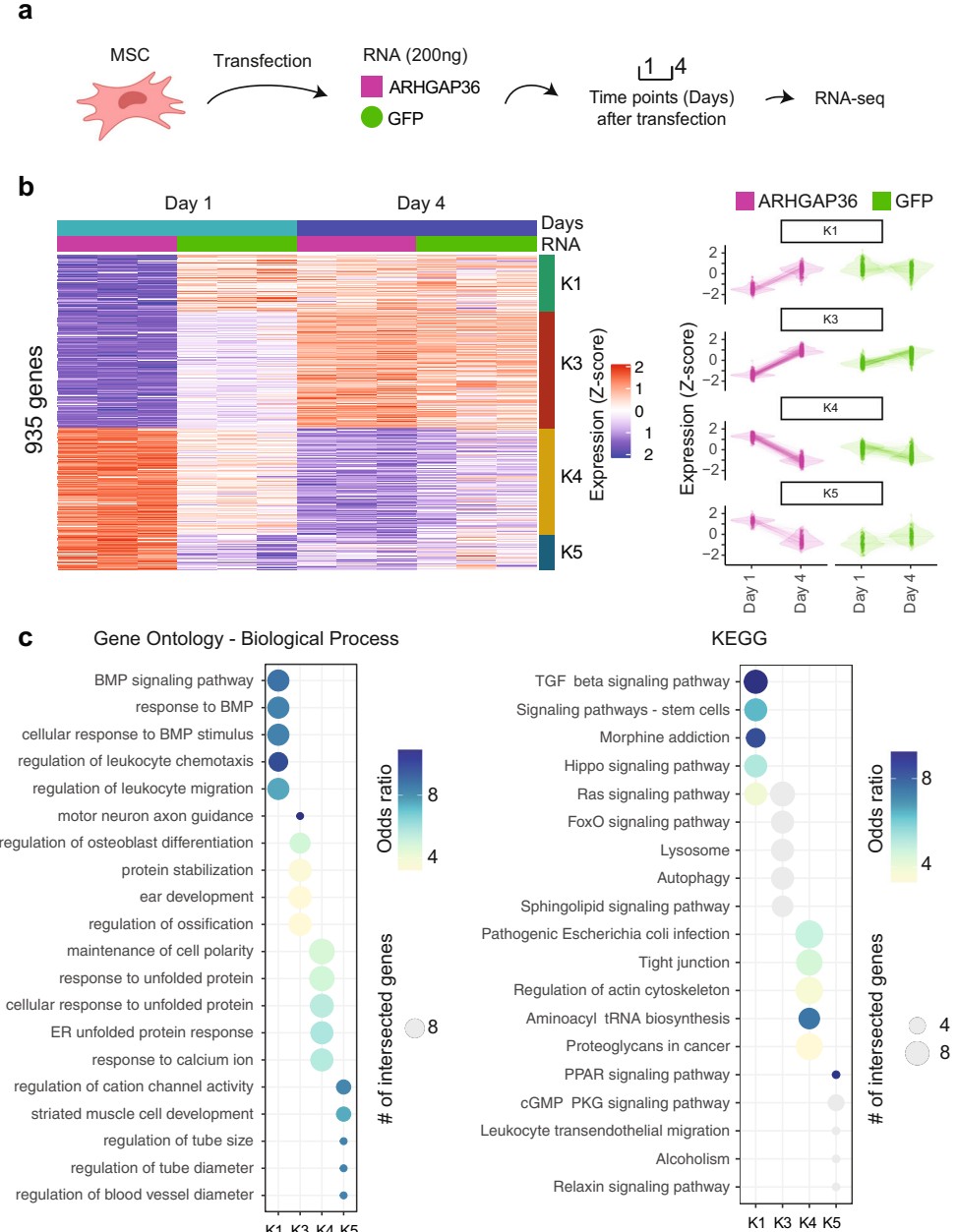

**Fig. 4 | RNA-seq in MSCs upon *ARHGAP36* and *GFP* transfection. a** Schematic representation of *ARHGAP36* and *GFP* transfection in MSCs. RNA-seq was performed in samples collected at 1 and 4 days after transfection. **b** Four clusters (K1, K3, K4, and K5) show high variability in ARHGAP36-transfect samples and minor variance among GFP ones. **c** Gene ontology and KEGG analysis revealed enrichment for TGFβ-BMP pathways in K1, regulation of osteogenesis in K3, and tight junction (cell-adhesion) in K4.

## Osteogenic differentiation occurs faster in proband cells and is independent of the HH pathway

in vitro, osteogenic differentiation of MSC and fibroblasts are a gold standard for studying bone formation in a dish[27,28]. Hence, we induced osteogenesis in fibroblasts from proband and controls for up to five weeks and tested for calcium deposition via alizarin red (AR) staining (Fig. 6a). The control values at each time point were set to 1.0, and the proband values were plotted as fold-change relative to the controls at the corresponding weekly time point. After two weeks of stimulation with an osteoinductive medium, proband cells showed increased in vitro calcium deposition compared to controls (Fig. 6b). The increase in calcium deposition reaches its peak after three weeks of differentiation, followed by a likely saturation of ECM production at week four. We then evaluated the influence of the HH pathway via *GLI1* expression during the osteogenic differentiation in these samples

(Fig. 6c). *GLI1* expression was reduced by 50% via the inhibitor GANT61, exposed for 48 h (Fig. 6d). GLI-blocking experiments revealed no difference in calcium deposition in both proband and controls in three different time points (Fig. 6d, e). We observed a reduction of roughly 10% of AR staining in all tested samples. We hypothesize that the molecular mechanism involved in the proband might be independent of the canonical HH pathway activity, as blocking *GLI1* is not sufficient to rescue the in vitro phenotype during osteogenic differentiation. Hence, crosstalk between several pathways such as HH, BMP-TGFβ and WNT, may result in synergistic effects in the proband (Fig. S10)[26,29–39].

## Discussion

Here we describe an ultra-rare condition presenting progressive ectopic calcification, in which various soft tissues of the proband were turned into the bone by an enhancer hijacking mechanism that

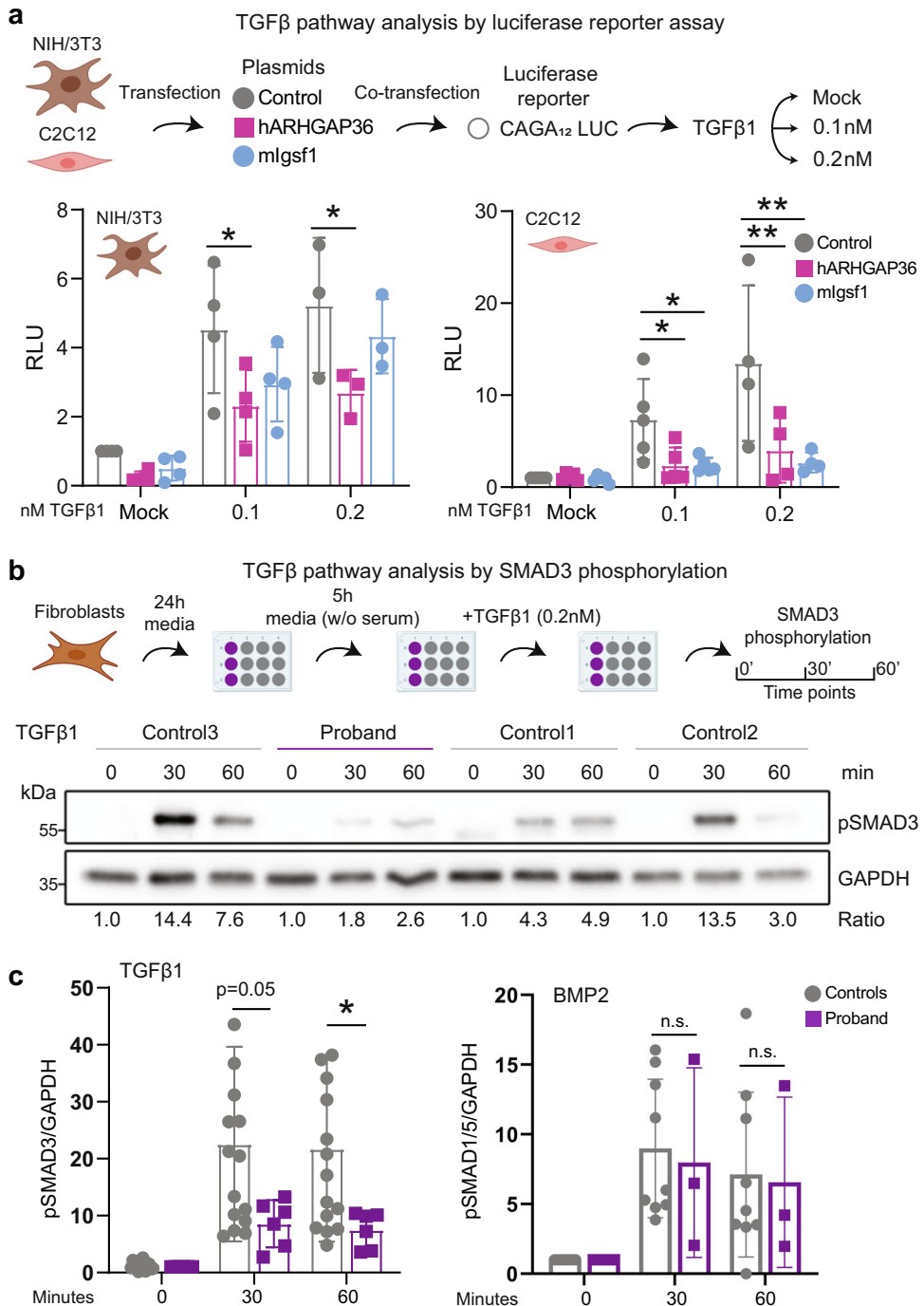

**Fig. 5 | *ARHGAP36* function in TGFβ pathway in mouse and human cell lines.**
**a** Schematics of transfection protocol in NIH/3T3 (fibroblast) and C2C12 (myoblast-like) mouse lines (top). Murine cells were transfected with an empty plasmid (as control), h*ARHGAP36* (purple), and m*Igsf1* (as positive control; blue) to evaluate TGFβ pathway activity. These cells were co-transfected with CAGA$_{12}$MLP-Luc plasmid (TGFβ-sensitive reporter). Two different TGFβ1 ligand concentrations were used in this assay (0.1 and 0.2 nM, plus MOCK). TGFβ1 in NIH/3T3 cells: Mock and 0.1 nM, $n=4$ technical replicates; TGFβ1 in C2C12 cells: Mock and 0.1 nM, $n=5$; 0.2 nM, $n=4$ technical replicates. NIH/3T3 cells showed a reduction of TGFβ activity after TGFβ1 induction at two concentrations (bottom). TGFβ inhibition is more dramatic in C2C12 cells at higher TGFβ1 concentrations. m*Igsf1*, here used as a positive control, only inhibits TGFβ in myoblast-like cells. Statistical significance within the indicated groups was calculated using two-way ANOVA and Dunnett's multiple comparisons tests; $p$-value: *<0.05, **<0.01. Relative Luminescence Units (RLU) are expressed as mean fold induction ±SD over unstimulated transfected control cells. **b** Schematic representation of the TGFβ experiment in proband fibroblasts and controls (top). Cells were seeded for 24 h in fibroblast media; on the next day, media was replaced by media without serum for 5 h. Cells were induced with TGFβ1 at 0.2 nM and collected at three-time points for SMAD3 phosphorylation analysis. **c** Proband cells show decreased pSMAD3 levels by western blot at 30 and 60 min after TGFβ1 induction (bottom) ($n=1$ proband, $n=3$ controls; 6 technical replicates of each sample). Densitometric quantification of pSMAD3 and pSMAD1/5 relative to GAPDH levels expressed as mean fold induction ±SD in arbitrary units. Source data are provided as a Source Data file.

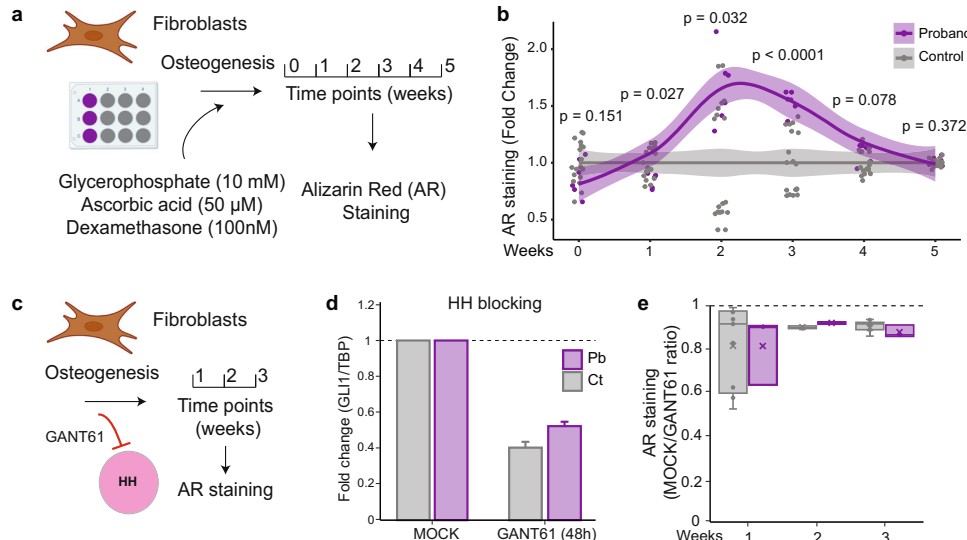

**Fig. 6 | Osteogenic differentiation in fibroblasts from the patient and controls.** **a** Schematic representation of the osteogenic protocol. Cells ($n = 1$ proband, $n = 3$ controls; 3 technical replicates of each sample) were stimulated with glycerophosphate (10 mM), ascorbic acid (50 μM), and dexamethasone (100 nM), were collected in six-time points (0–5 weeks) and stained for alizarin red (AR) for calcium deposition. **b** Patient samples showed a faster osteogenic differentiation after two weeks of stimulation, lasting 3-4 weeks. The control values at each time point were set to 1.0. The translucent band shows the confidence interval for each replicate for the respective time point. **c** Schematic representation of HH inhibition by GANT61 (1 and 5 μM) during the osteogenic differentiation. **d** GANT61 (5 μM) reduces by half *GLI1* expression in both patient and control samples after 48 h of molecule exposure ($n = 1$ proband, $n = 1$ control). Pb: proband; Ct: controls. RT-qPCR quantification is expressed as mean fold ±SD in arbitrary units. Source data are provided as a Source Data file. **e** HH inhibition by GANT61 (5 μM) revealed a minor decrease of 10% of AR staining in all samples. The median score is represented by the horizontal line in the center. The 25th and 75th percentile values are indicated by the lower and upper limits of the box. Source data are provided as a Source Data file.

activated *ARHGAP36*. The genetic cause was an inter-chromosomal insertion that reshuffled the 3D chromatin architecture at the *ARHGAP36* locus and had a significant impact on gene expression of bone-related signaling pathways.

FOP, the most studied HO disease by far, is caused by mutations in *ACVR1*, a gene encoding for a BMP receptor that induces osteogenic differentiation. In FOP, ectopic calcifications appear in the first decade of life[40], caused by increased SMAD1/5 phosphorylation downstream of the BMP signaling pathway[41]. The most frequent FOP mutation (ACVR1[R206H]) modifies important pathways in response to injury, causing a chronic pro-inflammatory state and abnormal skeletal muscle repair in patients[42,43]. Consequently, in FOP, the transition of tissue-resident fibroblasts to early chondrocytes and further to hypertrophic chondrocytes causes endochondral ossification instead of muscle repair[1]. In this study, we describe a HO phenotype that occurred spontaneously, without external triggers, characterized by a congenital rapidly progressive calcifications in the joints, diagnosed at the age of 5 months. The ectopic calcifications further progressed into the muscle of the jaw, hips, pelvis, shoulders, and limbs, and later on to general skeletal muscle until her death at the age of 8[13]. This unique and aggressive phenotype led us to investigate in greater detail the molecular mechanism causing the disease.

The 820 kb duplication from chr2 piggybacked six protein-coding genes and several regulatory elements to chrX in an intergenic region between *ARHGAP36* and *IGSF1*. First, we discarded the ORF-duplicated genes as causative for the disease (details in Supplementary Information). Next, we showed that the duplicated chr2 region had signals of chromatin activity in wild-type MSCs and fibroblasts (Fig. S2c). Thus, we analyzed the RNA-seq dataset as a proxy for the chromatin state of the der(X). Three genes, *ANXA4*, *GMCL1*, and *SNRNP27*, located within a novel putative chromatin domain (TAD #2), were expressed in both MSCs and fibroblasts wild type samples, but unexpectedly, they were silenced on der(X), even if duplicated. *IGSF1*, an important inhibitor of the TGFβ pathway in the testis and the pituitary gland[44], and therefore a suitable candidate to explain the disease, showed no expression in

both proband and controls. These data suggest that the chr2 active domain containing *ANXA4*, *GMCL1* and *SNRNP27* became inactive as a consequence of its insertion on der(X). On the other hand, the chr2 TAD containing the *GFPT1*, *NFU1*, and *AAK1* genes kept its chromatin activity considering their upregulation in proband fibroblasts. Lastly, in the Hi-C custom map, we observed a novel chromatin domain (Shuffled-TAD) at the left breakpoint (TAD #1), connecting *ARHGAP36* to regulatory elements located at the *ANTXR1* gene body (Fig. 2c, d). Interestingly, *ARHGAP36* is the gene with highest fold-change in the fibroblast RNA-seq dataset, suggesting this gene was likely activated in proband fibroblasts by enhancer hijacking.

ARHGAP36 is a poorly studied protein member of the RhoGT-Pases family, and its function in health and disease remains largely unknown. Previous studies have shown that ARHGAP36 is an agonist of the non-canonical HH signaling pathway and an antagonist of PKA signaling[26,29,45]. In the absence of HH ligands, *SUFU* represses the transcription factor Gli1 thereby inhibiting the HH pathway[26,45]. Its function in the non-canonical HH signaling pathway is mediated through promoting PKA degradation, subsequently leading to the activation of Gli transcription factors[26,29,46,47]. Using the RNA-seq data generated in this work, we evaluated the influence of *ARHGAP36* in the HH signaling pathway. We observed downregulation of *GLI3*, but *GLI1* and *CUL1* overexpression in the proband fibroblasts, all known markers of HH activation[48]. However, we cogitated that the HH signaling pathway alone might not be responsible for the extreme HO disease studied here since HH pathway inhibition in proband-differentiated osteoblasts did not cause changes in calcium deposition. Our data suggest that the HH pathway, coupled with other factors/pathways, may play a synergetic role in this disease.

Signaling pathways such as BMP-TGFβ, Notch, and WNT are known to be involved in the osteogenic process in health and disease[49–51], and HH and TGFβ signaling have overlapping synergistic effects in bone formation via Gli1/2[52–54]. In the fibroblast's RNA-seq data, the BMP-TGFβ and WNT pathways showed enrichment of DEGs in the proband sample. It is important to note that most of the WNT downregulated genes are

upstream inhibitors of this pathway, i.e., WNT signaling is active in the proband fibroblasts. Previous studies in osteoblasts have shown that PKA signaling activates WNT through inhibition of GSK3β, a known inhibitor protein of WNT signaling[30,55]. Here we hypothesized that *ARHGAP36* overexpression in the proband plays a role in several significant pathways related to bone formation. Therefore we validated these results via orthogonal in vitro experiments.

To overcome the genetic background potential bias from the proband fibroblasts, we transiently overexpressed *ARHGAP36* in MSCs and evaluated co-expression clusters (K1 to K10) showing variation in *ARHGAP36* transfected cells (at day 1 or 4) in comparison to *GFP*. Interestingly, the K1 co-expression cluster contains genes enriched for the TGFβ pathway in skeletal development[56]. Furthermore, another co-expression cluster (K3) showed several TGF-β target genes being downregulated in *ARGHAP36* samples (Fig. S6). For instance, the death-associated protein kinase (*DAPK1*) is involved in TGFβ dependent apoptosis, where its activation is mediated by SMAD2/3[57], and the selenoprotein P (*SELENOP*), which role is related to chondrocyte hypertrophy during development[58]. Indeed, orthogonal experiments overexpressing *ARHGAP36* in murine cell lines exposed to TGFβ1 showed reduced gene-reporter activity when compared to control. It is important to note that TGFβ1 strongly enhances bone formation induced by BMP2, indicating an important connection between TGFβ and BMP signaling in osteoblast differentiation[38]. This effect was more prominent in muscle-like cells (C2C12) than in fibroblast-like NIH/3T3 cells, suggesting that the genomic background plus the transcription factors related to specific cell types are important factors to be considered in this disease. Moreover, TGFβ1 exposure to fibroblasts slightly reduced TGFβ activity in the proband cells when compared to controls. Further, we observed independent TGFβ1-stimulation and enhanced baseline phosphorylation of p38 in proband cells. As p38 plays a pivotal role in different steps of osteoblast differentiation, mainly through the induction of pro-osteogenic transcription factors like RUNX2[59,60], it remains an interesting downstream effector of ARHGAP36-dependent enhanced pro-osteogenic capacities. While p38 activation can be induced by a variety of upstream triggers, including different biochemical and mechanical signaling cascades and cellular stresses, future studies should evaluate the link between ARHGAP36 function and p38 activation. Interestingly, extracellular matrix production directly modulates the WNT signaling pathway and other pathways involved in bone formation[61].

In summary, we identified novel−and validated known−functions of *ARHGAP36*, relating the expression of this gene to a severe case of HO. Moreover, these results merit a further functional exploration of this gene as a potential player in other connective tissue-to-bone formation diseases.

## Methods

### Samples collection and ethics committee approval
Healthy parents provided written informed consent on behalf of the female proband in this study. Blood collection for DNA screening and skin biopsy for fibroblast culture was performed after obtaining written consent from the parents. The study adhered to the Declaration of Helsinki standards and was approved by the internal Ethics Committee of the Department of Medical Sciences, University of Torino, Italy, under protocol number 0053916. Written informed consent was obtained for the collection of tissue samples for isolation human fibroblasts, mesenchymal stromal cells and chondrocytes, and ethics approval was obtained from the local ethics committee/institutional review board of Charité-Universitätsmedizin Berlin (EA2/089/20) and the province of Salzburg, Austria (415-E/1776/4-2014).

### Cell culture and samples
Fibroblasts were grown in DMEM supplemented with 10% fetal bovine serum (FBS), 1% L-glutamine, and 1% penicillin−streptomycin. Age-matched fibroblasts from three unrelated healthy female individuals were used as controls. NIH/3T3 and C2C12 cells were maintained and expanded in the media described above.

Mesenchymal stromal cells (MSCs), $n = 2$ male and $n = 1$ female, were isolated from human bone marrow from three donors and their phenotype was confirmed as previously described[62,63]. Mesenchymal stromal cells (MSCs) and chondrocytes were cultured in DMEM low glucose (Sigma-Aldrich) containing 10% human platelet lysate (pHPL), 1% L-glutamine and 1% penicillin−streptomycin.

### Plasmids
ARHGAP36 isoform 2 (ENST00000412432.6; UniProt ID: Q6ZRI8-2) plasmid was used for murine cell lines experiments. Full length mouse Igsf1 cDNA (ENSMUST00000033442.14; UniProt ID: Q7TQA1-1) was synthesized via GENEWIZ (Leipzig, Germany), PCR amplified using Phusion High-Fidelity DNA Polymerase (NEB) and subsequently purified, restriction digested and cloned into pcDNA3.1(-)myc-his (Thermo Fisher Scientific). Primer sequences for mIgsf1 cloning are: Forward: CTCGAGCGGCCGCgccaccatgatgcttcggaccttcactc; Reverse: GGGCCCAAGCTTtattggaactgtcagttccactgag.

In order to transiently express *ARHGAP36* and *GFP* in vitro transcribed mRNAs in MSC, the ARHGAP36 sequence was codon-optimized via the GeneArt Online tool (Thermo Fisher Scientific) for efficient overexpression. In vitro transcription was performed using different sets of nucleotides containing ATP, 5-methyl-CTP, GTP, 5-methoxy-UTP, and +ARCA cap analog. The plasmid vector pRNA2-(A) was used as a template for in vitro transcription of mRNA coding GFP[64].

IBMP response element reporter construct (BRE$_2$-Luc)[65] or a TGFβ response element reporter construct (CAGA$_{12}$-Luc)[66] were used in this work.

### Western blot and antibodies
Antibodies used in this study for western blot analysis were diluted (Primary; 1:1000; Secondary, 1:10.000) in 3% w/v bovine serum albumin (BSA)/fraction V (Carl Roth) in TBST (0.1% Tween). The following antibodies were used: ARHGAP36 (HPA002064, Atlas Antibodies); GFP (2956, Cell Signaling Technologies); phosphorylated SMAD1/5 (Ser463/465), clone 41D10 (Cell Signaling Technologies); phosphorylated SMAD3 (Ser423/425), clone C25A9 (Cell Signaling Technologies); SMAD1, clone D59D7 (Cell Signaling Technologies); SMAD3, clone C67H9 (Cell Signaling Technologies); SMAD2/3, clone D7G7 (Cell Signaling Technologies); GAPDH, clone 14C10 (Cell Signaling Technologies); phosphorylated PKA substrate (RRXS/T) 100G7 Lot.:4 rabbit mAb 9624S (Cell Signaling Technologies); phosphorylated TAK1 (Ser412) #9339 (Cell Signaling Technologies); phospho-p38 MAPK (Thr180/Tyr182) (28B10) Mouse mAb #9216 (Cell Signaling Technologies); RUNX2 (6H4L27, Thermo Fisher Scientific); COL1A1 (E3E1X, Cell Signaling Technologies); and COLX (JF0961, Thermo Fisher Scientific).

Protein lysates were subjected to SDS-PAGE and transferred to PVDF membranes by western blotting. Membranes were blocked for 1 h in 0.1% TBS-T containing 3% w/v BSA, washed three times in 0.1% TBS-T, and incubated with indicated primary antibodies overnight at 4 °C. For HRP-based detection, goat-α-mouse or goat-α-rabbit IgG HRP conjugates (Dianova) were used. Chemiluminescent reactions were processed using WesternBright Quantum HRP substrate (Advansta) and documented on a FUSION FX7 digital imaging system.

### Cytogenetics, transcriptomic, and genomics analyses
Fibroblast total RNA from the proband and three controls were extracted using RNeasy Mini Kit (Qiagen), and MSCs RNA was extracted using TRIzol Reagent (Thermo Fisher Scientific).

RT-qPCR was performed using the PowerUp SYBR Green Master Mix (Thermo Fisher Scientific), submitted to the QuantStudio 6 System (Applied Biosystems). *GLI1* expression was calculated using the $2^{-\Delta\Delta CT}$ method[67], and *TBP* was used as a normalizer. Each experiment was

performed with three technical replicates. Protein lysates were extracted using RIPA buffer (150 mM NaCl, 50 mM Tris, 0.1% SDS, 1% NP-40 Alternative).

Exome (ES) and genome sequencing (GS) were performed using the genomic DNA of the trio (average depth 30×) on Illumina HiSeq X machines with Illumina TruSeq PCR-free chemistry. Sanger sequencing was performed to map the breakpoints at the base pair level. Primer sequences are: (1) BP1-Forward: GCTAATGAATTTCAACCCTGG; BP1-Rreverse: GAAGATTCAAAGCCGAATGG; (2) BP2-Forward: GCTGCAGG ACAGTCACAAGG; BP2-Reverse: GTCAGAGTCGCTCACACTGC. Break-points identified by inverse PCR have been validated on the proband's genomic DNA using: (3) BP1-ChrX-Forward: CCTTCACATCCCTT GTAAGTTG; BP1-Chr2-Reverse: TTGGACAGGCTGAACAGTGG; (4) BP2-Chr2-Forward: TCCCTGTTGGTTCTGATTAGG; BP2-ChrX-Reverse: GGGAAGTAAAGCTCTCCTCAGC.

Array-based comparative genomic hybridization (Array-CGH) was performed using the Agilent Human Genome Microarray Kit 244K (Agilent Technologies). The detected duplication/insertion was detected by trio fluorescence in situ hybridization (FISH) metaphases using BAC probes overlapping the *MXD1* gene, further confirmed by fiber-FISH on metaphases of the proband.

Hi-C libraries were performed using a protocol described elsewhere[68], with minor adaptions[69]. The DNA was prepared for Illumina short-reads sequencing by ligating adapters to the DNA fragments using the NEBNext Multiplex Oligos kit and amplified by PCR. Four libraries per case were sequenced for 100 M fragments each, PE-100 bp on a HiSeq4000 (Illumina).

RNA-seq was performed in fibroblasts and MSCs using the poly(A) mRNA capture and the KAPA mRNA HyperPrep Kit (KR1352) in three technical replicates. Libraries were sequenced on a HiSeq4000 (Illumina; PE-75 bp), with ~50 million fragments per sample.

### Bioinformatics analyses

**Exome and genome sequencing analyses.** Illumina sequencing reads were mapped to the GRCh37/hg19 genome of reference with BWA-MEM[70] and variants were called using GATK[71]. Using trio-genome sequencing (GS) data, the proband haplotypes were phased using the GATK HaplotypeCaller[72]. We developed an allele-of-origin prediction tool based on the number of phased-variants per read, comparing to the reference annotation. Reads carrying variants with known maternal or paternal inheritance are clustered together and the read coverage of each cluster supports the parent of origin for the target-duplicated region.

**RNA-seq.** RNA-seq PE reads were mapped to the human genome build hs37d5 using STAR[73], and gene expression was retrieved using DESeq[74]. We consider differentially expressed genes (DEG) those that show adjusted $p$-value < 0.001 and an absolute $\log_2$ fold change > 2. X chromosome inactivation was evaluated by using GS-phased variants and RNA-seq expression data to detect allele-specific gene expression.

**Hi–C.** Hi–C paired-end sequencing data were processed using the Juicer pipeline[75], and the bioinformatics pipeline is detailed elsewhere[69]. We used one female fibroblast Hi-C map as a control. We generated a customized genome based on hg19 to reflect the duplication and insertion detected in the proband and repeated the data processing with the Juicer pipeline described above. In the customized genome, chrX was replaced by der(X), and the original sequence of the duplicated region on chr2 was masked to allow the mapping of short reads to the chimeric part of der(X). Note, as a consequence, the Hi–C reads from the original region on chr2 are also mapped to der(X), creating an overlay of the Hi–C signal. Genome-wide Hi–C maps were visualized using Juicebox[75], and for the visualization of interchromosomal maps for the locus of interest, we used an in-house program and the Hi-C maps are displayed as heatmaps rotated by 45°.

Values above the top 99.2th percentile were truncated to improve the display of smaller count values.

**Enhancer prediction.** We used the condition-specific regulatory units prediction tool (CRUP)[76] for enhancers prediction in MSC and fibroblasts using the ChIP-seq epigenetic dataset containing H3K27ac, H3K4me1, and H3K4me3 marks, described elsewhere (Hochmann et al.[77]). CRUP uses the information of the three above-mentioned histone marks to calculate the probability that a given region in the genome harbors an enhancer element.

### In vitro experiments

**Human fibroblasts.** Fibroblasts were induced to osteogenic tissue using a protocol detailed elsewhere[78]. In short, $8 \times 10^4$ cells (one proband and three controls; three biological replicates per sample) were seeded onto a 12-well plate. After 24 h, cells were stimulated with osteogenic media (OM) containing DMEM low glucose, 10% FCS, 1% pen/strep, glycerophosphate (10 mM), ascorbic acid (50 μM), and dexamethasone (100 nM) over five weeks. We added Ficoll 70 (37.5 mg/ml) and Ficoll 400 (25 mg/ml) to the OM to avoid cell detachment during differentiation. Cells were fixed in 4% PFA at six different time points (0–5 weeks) for alizarin red (AR) staining. Samples were measured on Infinite® 200 PRO (wavelength 562 nm; two technical replicates per biological replicate).

**Cell stimulation with growth factors.** To evaluate the BMP and TGFβ pathways, we stimulated fibroblasts with rhBMP2, rhTGFβ1 (Pepro-Tech) or rhActivinA (R&D Systems) reconstituted in PBS for three time points (three biological replicates per sample). At day 1, $2 \times 10^5$ fibroblasts were seeded in 6-well plates in 2 mL DMEM, 10% FCS, 1% L-Glutamine, and 1% pen/strep. The next day, media was changed to starvation media (DMEM without serum, 1% L-Glutamine and 1% pen/strep) and cells were starved for five hours. Next, cells were stimulated with BMP2 (5 nM), TGFβ1 (0.2 nM) or Activin (3 nM) in PBS for 0, 30, and 60 min. After stimulation, cells were washed with 1× PBS and added 1× SDS Laemmle buffer. Scraped protein lysates were boiled at 95 °C for 5 min and froze at −20 °C. At day 3, we analyzed SMAD1/5 and SMAD2/3 phosphorylation via western blot.

**Mouse cell culture and dual luciferase reporter gene assay.** NIH/3T3 and C2C12 cells ($1.5 \times 10^4$) were seeded for luciferase reporter gene assay in a 96-well plate. BRE$_2$-Luc or the CAGA$_{12}$-Luc plasmids were transfected together with pcDNA.1/myc-His(−) empty vector (Thermo Fisher Scientific), mIgsf1-myc-His or isoform 2 ARHGAP36-N-mCherry[25] using Lipofectamine 2000 (Thermo Fisher Scientific) according to manufacturer's instructions. A constitutively expressing construct encoding renilla luciferase (RL-TK, Promega) was co-transfected as internal control. The next day, cells were starved in serum-free DMEM for four hours and stimulated with BMP2 or TGFβ1 overnight. Cell lysis was performed using passive lysis buffer (Promega) and measurement of luciferase activity was carried out according to the manufacturer's instructions using a TECAN infinite f200 Luminometer. Data are shown as relative light units (RLU) normalized to the empty vector control. The experiments were performed with $n = 3$, 4, or 5 technical replicates, as stated in respective figure legends.

**MSCs and chondrocytes transiently overexpressing *ARHGAP36*.** MSCs from one healthy donor were seeded in a 48-well plate in MSC media for 24 h in triplicate. ARHGAP36-optimized codon and GFP in vitro transcription (IVT)-mRNA were produced using TranscriptAid T7 High Yield Transcription Kit (Thermo Fisher Scientific) according to the manufacturer's instructions. On the next day, mRNA was transiently transfected at 200 ng with 0.3 μl Lipofectamine MessengerMax per well (Thermo Fisher Scientific). Cells were collected at 1 and 4 days after the transfection for RNA-seq.

MSCs and chondrocytes were seeded in a 48-well plate in MSC media for 24 h in triplicate. In the next day, the cells were exposed to OM for up to 14 days. One day after OM stimulation, MSCs and chondrocytes were transiently transfected with ARHGAP36-optimized codon sequence and GFP plasmids. Plasmids retransfection was performed after 7 days and cells were collected at different time points for Western blot.

## Statistical analysis

Statistical analyses of densitometric protein level quantification, dual luciferase assay, and quantitative image analysis were performed using GraphPad Prism 8. Two-way ANOVA with Bonferroni's or Dunnett's multiple comparisons post hoc test were used to check for statistical significance, respectively. For all experiments, statistical significance was assigned with an alpha-level of $p < 0.05$.

## Additional software

We used biorender to create schematic representations of Figs. 3–6 and Figs. S2, S4–S9. STAR
DESeq: Juicer and Juicebox: http://aidenlab.org/documentation.html, HaplotypeCaller (V:3.2-2-gec30cee):
BWA-MEM (0.7.10-r789): GATK (3.4-46-gbc02625).

## Reporting summary

Further information on research design is available in the Nature Portfolio Reporting Summary linked to this article.

## Data availability

Informed consent in this work does not cover the deposition of sequencing data from the proband sample. The whole genome sequencing data and Hi-C generated in this study are available under restricted access for patient privacy; access can be only obtained by request via data use agreements, which must be signed by our group and the applicant University/Research Institute. The processed sequencing data are available upon request from M.Spielmann (malte.Spielmann@uksh.de) with the period for respond to the access request of one calendar month. The processed data cannot be shared with third parties; if the data will be used for scientific presentations and/or publications, the applicant should contact M.Spielmann for further details. The raw sequencing data are protected and are not available due to data privacy laws. Data can only be shared for research purposes with permission of the patient's legal guardian(s). All other data supporting the findings described in this manuscript are available in the article, Supplementary Information, and source data file. Source data are provided with this paper.

## Code availability

Code for haplotyping and RNA-seq phasing, and X-inactivation analysis:https://github.com/moeinzadeh/parent-of-origin.

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

## Acknowledgements

We would like to thank the family for their contribution to this project, remembering the love and joy that have been donated to the proband. We would like to thank Oliver Rocks, Institute of Biochemistry—Charité, Germany, for the courtesy of the ARHGAP36 plasmid and Walter Sebald, University of Würzburg, Germany, for the rhBMP2. The authors would also like to thank the BCRT Cell Harvesting Core Unit (BCRT-CH) of the Berlin Institute of Health, Charité—Universitätsmedizin Berlin, for their excellent technical assistance and support. This study was supported by the German Federal Ministry of Education and Research (BMBF, 031L0234B), the European Union's Horizon 2020 research and innovation program (Grant No. 779293), and the German Research Foundation (DFG) through funding of the Research Group 2165 (GE2512/2-2), (SFB1444) to P.K. and the Collaborative Research Center 1444. Genome sequencing was provided by the University of Washington Center for Mendelian Genomics (UW-CMG) and was funded by NHGRI and NHLBI grants UM1 HG006493 and U24 HG008956. This study was also supported by Ministero dell'Istruzione, dell'Università e della Ricerca—MIUR "Dipartimenti di Eccellenza 2018–2022" to Department of Medical Sciences, University of Torino (Project no. D15D18000410001), and Italian Ministry of Health (5 × 1000), Fondazione Bambino Gesù (Vite Coraggiose) to M.T.

## Author contributions

U.S.M., J.J., E.G., and M. Spielmann conceived the study. U.S.M., J.J., S.H., P.VG., S.A., and M.-K.K. performed cell culture experiments using fibroblasts, NIH/3T3 and C2C12 cells, DNA- and RNA-library preparations. U.S.M., J.J., S.H., P.V-G., M-J.O. and M.Schwetska performed experiments using MSCs, RNA-seq and Western blot. U.S.M and M.-K.K. performed Hi–C experiments. U.S.M. and S.H. analyzed customized Hi–C maps. C.A.P-M., R.S., M.-H.M., and A.A. performed all bioinformatics analyses. M.-H.M. performed Illumina genome sequencing read alignment and SV calling. R.S. processed Hi–C data, created chromosome reconstructions, and derived Hi–C maps. M.-H.M. performed haplotyping of patient genomes and phasing of RNA-seq data. C.A.P.-M. performed differential gene expression analysis. E.F., L.B., A.C., G.C., M.D., G.B.F., M.T., A.B., M.G., D.S., S.G., S.M., S.S., and P.K., contributed with sample collection and clinical evaluation. U.S.M., J.J., and S.H. wrote the paper. S.S., P.K., E.G., and M. Spielmann reviewed the paper.

## Funding

## Competing interests

The authors declare no competing interests.

## Additional information

[1]Max Planck Institute for Molecular Genetics, Development and Disease Group, 14195 Berlin, Germany. [2]Institute for Medical Genetics and Human Genetics, Charité University Medicine Berlin, 13353 Berlin, Germany. [3]Freie Universität Berlin, Institute for Chemistry and Biochemistry, 14195 Berlin, Germany. [4]Istituto Superiore di Sanità, Department of Oncology and Molecular Medicine, 00161 Rome, Italy. [5]Cytogenetics Unit, Casa Sollievo della Sofferenza Foundation, IRCCS, 71013 San Giovanni Rotondo, Foggia, Italy. [6]Molecular Genetics and Functional Genomics, Ospedale Pediatrico Bambino Gesù, IRCCS, 00146 Rome, Italy. [7]Max Planck Institute for Molecular Genetics, Department of Computational Molecular Biology, 14195 Berlin, Germany. [8]Department of Biotechnology, University of Verona, 37129 Verona, Italy. [9]Julius Wolff Institute (JWI), Berlin Institute of Health at Charité – Universitätsmedizin Berlin, 13353 Berlin, Germany. [10]BIH Center for Regenerative Therapies (BCRT), Berlin Institute of Health at Charité – Universitätsmedizin Berlin, 10117 Berlin, Germany. [11]Institute of Active Polymers, Helmholtz-Zentrum Hereon, 14513 Teltow, Germany. [12]Berlin-Brandenburg Center for Regenerative Therapies (BCRT), 13353 Berlin, Germany. [13]Department of Clinical and Biological Sciences, University of Torino, 10043 Torino, Italy. [14]Department of Medical Sciences, University of Torino, 10126 Torino, Italy. [15]Medical Genetics Unit, Città della Salute e della Scienza University Hospital, Torino 10126, Italy. [16]Cell Therapy Institute, Spinal Cord Injury and Tissue Regeneration Center Salzburg (SCI-TReCS), Paracelsus Medical University (PMU), 5020 Salzburg, Austria. [17]Department of Molecular Medicine, University of Pavia, 27100 Pavia, Italy. [18]Medical Genetics Unit, IRCCS Mondino Foundation, 27100 Pavia, Italy. [19]Institute of Human Genetics, University Hospitals Schleswig-Holstein, University of Lübeck and University of Kiel, Lübeck 23562, Germany. [20]DZHK (German Centre for Cardiovascular Research) Germany, partner site Hamburg, Lübeck, Kiel, Lübeck 23562, Germany. ✉e-mail: umelo@molgen.mpg.de; elisa.giorgio@unipv.it; malte.Spielmann@uksh.de

