## [Peer Review File · Nature Communications]

Enhancer hijacking at the ARHGAP36 locus is associated with connective tissue to bone transformationREVIEWER COMMENTS

Reviewer #1 (Remarks to the Author):

This is a very interesting and solidly performed piece of work, which is interesting from multiple angles, gene regulation, clinical genetics, mutational mechanism, and ossification biology. They show that the genetics in this unfortunate individual is due to a de novo duplication of an 820kb gene-rich region from chr2 onto ChrX. They stratify the likely causal genes initially on the basis of function but confirm the likely candidate by both gene expression analysis (RNA-seq) and functional overexpression analysis in cellular models and analyses the common affected pathways. The proposed mechanism of "enhancer hijacking" is supported by Hi-C analysis and ChIP-seq based enhancer identification. In general, I find the story compelling and of very good general interest, though I would say that the "enhancer hijacking" is the most likely mechanism rather than completely proven. This may be being somewhat completist in these regards but it is the genomic mechanism that I personally find the most interesting.

1. I would like to have seen a follow-up with a higher resolution targeted 3C method to show conclusively that the promoter of the ARHGAP36 gene actually makes close contacts with some of the highlighted enhancers and if so with which ones. Maybe some virtual 4C using the HI-C data from the viewpoint of the enhancers or promoter would be helpful here to firm up the mechanistic hypothesis.
2. I would have loved to have seen enhancer deletion analysis on the probands cells. This may not have been logistically possible (it is a duplication after all) but in the absence of this and the high resolution 3C, "enhancer highjacking" while it remains to most likely hypothesis doesn't feel fully nailed down to me, so it could be more appropriate to reflect that in the manuscript and back off from the complete certainty with which the state this as the mechanism.
3. How certain do they feel they really know the complete structure of the duplicated locus? It is derived only from short reads and junction analysis. Short-read sequencing is often very poor in detecting structural rearrangements, which can arise as a secondary event to the initial recombination. The Hi-C data could certainly help here but as it was described I was left a little unsure that it was just an assumption and it would be good if the authors expanded on that in the methods or text.

In general a very interesting and well-performed piece of work, which I personally found very interesting and will undoubtedly be of interest to a wide range of readers.

Reviewer #2 (Remarks to the Author):

The manuscript by Melo et al. reports on a patient with a severe and rare form of heterotopic ossification (HO). The patient had an duplication of ~820 kb from chr2 inserted into chrX. This led to alterations of a TAD and misexpression of a gene on chrX, ARHGAP36. ARHGAP36 is known to inhibit PKA signaling and activate Hedgehog (HH) signaling (although the mechanism(s) involved in the latter are not fully clear with some contradictory data in the literature). RNAseq analyses of control fibroblasts and fibroblasts from the proband, as well as of control and ARHGAP36 transfectants revealed changes to HH and TGFbeta pathway-associated genes. Overexpression of ARHGAP36 in fibroblasts and C2C12 myoblasts reduced TGFbeta signaling. Studies with fibroblasts from the proband and three controls were also studied for responses to TGFbeta, BMP2, and a GLI inhibitor, as well as for in vitro osteogenesis. The authors conclude that misexpression of ARHGAP36 leads to alterations in TGFbeta and HH signaling, which in turn underlie the clinical manifestation of the duplication/insertion.

The genomic analyses leading to the conclusion that ARHGAP36 is the gene critically affected by the duplication/insertion appear to be well performed, but this is not an area of expertise for me, so I defer to reviewers with greater experience. The RNAseq analyses showed changes to pathways that would be expected to be altered in cells with aberrant osteogenesis, and provided useful leads for the signaling studies (along with what is known of ARHGAP36 function). I find, however, that the signaling studies fall short of the level required for firm conclusions.

1. Figure 5: The studies with TGFbeta are not convincing. Although the TGFbeta response in cells overexpressing ARHGAP36, or in proband fibroblasts, trends lower than controls, in most cases it does not rise to the $p < 0.05$ value selected for significance. One major reason for this seems to be the extreme variability of the controls. This is seen even in panel B, where treatment of C2C12 cells with 0.2 nM BMP2 led to the greatest differential. The 3 controls in this panel differ from each other by 5- to 6-fold in an assay where the signal to noise (+/- ligand) averages 10-fold. These results are not compelling. In panel C, using proband-derived fibroblasts is a great approach, and I give the authors credit for using three different controls, but the controls are so different from each other in pSMAD3 levels, it is not possible to conclude anything. Similarly, pSMAD1/5 levels are extremely variable in both proband and controls.
2. Figure S9: This figure has a number of peculiar aspects to it. ARHGAP36 levels drop very quickly after transfection (reason not clear) so the cells were transfected again after 7 days in culture, yet this second large increase in ARHGAP36 levels had essentially no effect on target expression. This goes unmentioned. Additionally, although the fold-difference in expression of COL1A1, etc. is higher at day 1 with ARHGAP36, the overall levels at this time point are rather low, as all these target proteins are induced over the culture period. In fact, looking at the Western blots, it seems all the ARHGAP36 overexpression is doing is very modestly accelerating osteogenesis.
3. Figure 6: I don't understand panel B. Since the data are reported as fold-change in AR staining, it appears that the control cells did not change at all over 5 weeks (the value is 1 all the way across). In contrast, the proband cells show elevated AR staining at 2-3 weeks (as would be expected), but returns to a value similar to control at 5 weeks. How is this possible, that week 5 is lower than weeks 2-3? Panels D and E are a reasonable beginning to analysis of HH signaling, but use of GANT61 is not a sufficient approach to draw firm conclusions. Also, in panel D, the use of fold-change values obscures whether the proband cells have higher absolute levels of GLI1 expression than control cells – which would be expected if the authors hypotheses are correct.

Taken together, while I think the identification of the patient's duplication/insertion and the role of ARHGAP36 as the likely key gene are interesting, the mechanistic studies do not provide a compelling view of how this genome perturbation results in HO.

Reviewer #3 (Remarks to the Author):

The authors have conducted a thorough cytogenetics and genomics approach to discover HO-causing gene in a proband with progressive ectopic ossification. They demonstrated that interchromosomal insertional duplication of 820kb on chromosome 2 insertion at the ARHGAP36/IGSF1 locus in chromosome X resulted in enhancer hijacking thereby increasing expression of ARHGAP36 in proband's fibroblasts. Functional relevance of ARHGAP36 up-regulation in the severe HO case was investigated by overexpression studies using primary cellular systems in vitro. Several candidate signaling pathways identified by RNA-seq and DEG enrichment analysis were examined and, among others, TGFβ signaling was found to be marginally inhibited by ARHGAP36 overexpression. Interestingly, despite increased ECM deposition by ARHGAP36-overexpressing cells and accelerated osteogenesis of proband's fibroblasts, hedgehog signaling was not significantly influenced by ARHGAP36 overexpression. Identification of ARHGAP36 as a novel HO-causing gene is intriguing. However, the evidences for disease-causing functionality of ARHGAP36 presented in this study are not convincing enough to directly link it to HO.

Main Concerns

- (1) Any histology data with proband's biopsy, showing endochondral ossification vs. intramembranous ossification?
- (2) GTPase activity assay data of proband's ARHGAP36?
- (3) In addition to the luciferase assays for TGF β /BMP signaling, target gene expression of the TGF β /BMP signaling pathways can support the presented data.
- (4) Authors examined canonical TGF β signaling by looking at phos-SMAD3 levels. I am wondering if non-canonical pathways is influenced by ARHGAP36 overexpression, since TAK1 was known to negatively regulate HH-Gli pathway.
- (5) As authors speculated in Discussion, what if cells are co-stimulated with hedgehog and TGF β ? Would it bring out the phenotype caused by ARHGAP36 misexpression in proband's FB?
- (6) PKA downstream substrates were affected (e.g. phosphorylation) by proband's ARHGAP36?
- (7) Some minor suggestions: In addition to quant data of alizarin red stain for matrix calcification, images of ARS stained cells would be nice.

Reviewer #4 (Remarks to the Author):

The starting point of this topical pathophysiological study is a young patient with an ultra-rare condition characterized by a dramatic and progressive ectopic calcification, leading to early death due to heterotopic ossification, which is aberrant ossification of connective tissue.

Using extensive mechanistic studies, it has been shown that this condition is caused by an inter-chromosomal insertional duplication that disrupts a topologically associating domain of the X chromosome. This results in 'enhancer hijacking' and ARHGAP36 misexpression. ARHGAP36 activation has been shown to have an impact on gene expression of key pathways implicated in bone formation and heterotopic ossification.

The conclusions of the study are supported by extensive and well-conducted experimental work including functional studies on biologically relevant patient material, which is a strength. A potential limitation of the study is that it deals with one single case having an ultra-rare disease. It would be interesting to know if attempts have been done to find additional cases f.i. using GeneMatcher. This minor weakness is balanced by the excellent functional work supporting the proposed disease mechanism.

Overall, this is a unique study demonstrating that disruption of the 3D genome and enhancer hijacking caused by an inter-chromosomal duplication activates ARHGAP36, leading to soft tissue to bone transformation.

There are some minor points to address:

1/Abstract: The abbreviation TAD could already be introduced in the abstract

2/ Introduction: the references jump from 12 (Introduction) to 59 (Methods)

Methods

3/ IRB number of the research protocol?

4/ Cell culture and samples: is RNA total RNA?

5/ p.6: how has the specificity of the antibodies been assessed?

6/ array CGH, array-CGH: only abbreviation mentioned

7/ in situ hybridization: in situ in italics

8/ p.6: the title 'Cytogenetics and genomics' does not cover Hi-C and RT-qPCR

9/ RNA-seq and RNA-Seq

10/ p.7: Exome and Genome sequencing: already abbreviated

11/ Hochmann et al. 2022: status of the manuscript?

12/ In vitro osteogenic differentiation in proband cells is used as an interesting model for bone formation in a dish. Could be explained why induced fibroblasts are used, and why no reprogramming (iPSC) and differentiation is needed?

Results

13/ Abbreviations: osteogenic media (OM)(p.10): OM is already used before; heterotopic ossification is abbreviated (HO) but the abbreviation is not used on p.10 (title) and p.11 (text).

MRI: already abbreviated before?; p.10: abbreviation MSC used but only explained on p.12.

14/ functionally relevant genes are based on the clinical presentation: how is this defined?

15/ p.11: open reading frame instead of 'open read frame'

16/ p.12: The chr2 duplicated region partially maintainS

17/ p.13: we excluded skewED X chromosome inactivation.

18/ p.13: gene body: is the transcriptional unit meant?

19/ CRUP: abbreviated on p.8 and p.13

20/ p.13: differentiation via HH pathway: via 'the' HH pathway

21/ p.15: We observed *a* slightly reduced (...) levels: remove 'a'

22/ p.15: (...) although *we* did not reach significance: 'we' is missing

23/ p.15: Title: shows increased ECM deposition

Discussion

24/ p.16: by 'an' enhancer hijacking mechanism

25/ p.19: death associated protein kinase 1 (DAPK1)

Acknowledgements

26/ the name Beatrice is mentioned, assuming the parents consented with this.

Figures

27/ Figure 1: both Array-CGH and Array CGH used.

Supplementary Information

28/ Table S1 (several genes associated to heterotopic calcification/ossification disorders): mentioned on p.11 but could not be found in the Supplementary Information.

29/ Duplicated genes excluded as candidates to explain the phenotype. Although it is extensively motivated why several duplicated genes can be excluded as candidate genes for the phenotype, it should be mentioned that this is speculative. References are missing for some genes. Decipher with capital. Drosophila: to capitalize and italicize.

30/ p.3: skewED X inactivation; chromosome 2 instead of chromosomeS; showed AN increased number of maternal reads; de novo: italicize.

31/ p.5: Two regimes ARE observed

Reviewer #1 Comments:

This is a very interesting and solidly performed piece of work, which is interesting from multiple angles, gene regulation, clinical genetics, mutational mechanism, and ossification biology. They show that the genetics in this unfortunate individual is due to a de novo duplication of an 820kb gene-rich region from chr2 onto ChrX. They stratify the likely causal genes initially on the basis of function but confirm the likely candidate by both gene expression analysis (RNA-seq) and functional overexpression analysis in cellular models and analyses the common affected pathways. The proposed mechanism of “enhancer hijacking” is supported by Hi-C analysis and ChIP-seq based enhancer identification. In general, I find the story compelling and of very good general interest, though I would say that the “enhancer hijacking” is the most likely mechanism rather than completely proven. This may be being somewhat completist in these regards but it in the genomic mechanism that I personally find the most interesting.

R: We thank Reviewer #1 for the positive comments regarding our manuscript.

1. I would like to have seen a follow-up with a higher resolution targeted 3C method to show conclusively that the promoter of the ARHGAP36 gene actually makes close contacts with some the highlighted enhancers and if so with which ones. Maybe some virtual 4C using the HI-C data from the viewpoint of the enhancers or promoter would be helpful here to firm up the mechanistic hypothesis.

R: We have performed virtual 4C using ARHGAP36 promoter as viewpoint and we confirmed the results observed in Hi-C: gain of chromatin contact between ARHGAP36 promoter with putative enhancers located on the ANTXR1 gene body. Legend: CT - control; PB - proband.

2. I would have loved to have seen enhancer deletion analysis on the probands cells. This may not have been logistically possible (it is a duplication after all) but in the absence of this and the high resolution 3C, “enhancer highjacking” while it remains to most likely hypothesis doesn’t feel fully nailed down to me, so it could be more appropriate to reflect that in the manuscript and back off from the complete certainty with which the state this as the mechanism.

R: We agree with Reviewer #1 that the most likely pathomechanism involved in the studied disease is enhancer hijacking, yet not fully proven in this manuscript. Deletion of chr2 enhancers in the Shuffled-TAD would give a direct answer to support this hypothesis, however, as mentioned by Reviewer #1, the fact that the mutation is a duplication, makes it harder to delete only the Shuffled-TAD der(X) enhancers, without causing off-target in both chr2 wild-type alleles. Moreover, deleting the Shuffled-TAD enhancers could influence the X inactivation process, leading to unwanted results regarding this disease.

3. How certain do they feel they really know the complete structure of the duplicated locus? It is derived only from short reads and junction analysis. Short-read sequencing is often very poor in detecting structural rearrangements, which can arise as a secondary event to the initial recombination. The Hi-C data could certainly help here but as it was described I was left a little unsure that it was just an assumption and it would be good if the authors expanded on that in the methods or text.

R: We agree with Reviewer #1 that Illumina short-read sequencing presents some issues to pinpoint the correct breakpoint at base pair level in complex regions of the genome, a fact that could have been overcome by other technologies such as long-read technologies. The sensitivity of a Illumina short-read technology depends not only on the size of the structural

variants (SV), but also on the SV type (Chaisson et al. 2019; Ebert et al. 2021). We and others have shown that Hi-C coupled with short-read data is precise enough to: a) detect the breakpoints at high resolution; and b) is very helpful to reconstruct haplotypes and restructured 3D chromatin domains (Melo et al., 2020; Schopflin & Melo et al., 2022 - accepted for publication in Nature Communications). We are very confident that the here detected duplication is not part of a complex rearrangement. It was inserted on chrX and it is 820 kb long according to several technologies: a) array-CGH (Figure 1b; low resolution technique); b) Illumina short-read sequencing (Figure 1c; one of the breakpoints was mapped at the base pair level using Illumina short-read, and the second breakpoint with Sanger sequencing); and c) Hi-C (Figure S2a; 5kb resolution). Using Hi-C only (Figure S2a), one could detect the duplication within 5 to 10kb resolution (which would be roughly 810-830kb size). Furthermore, Hi-C clearly shows that no other rearrangement occurred neither at the duplicated locus nor at the insertion breakpoint (Figure S2a).

In general a very interesting and well-performed piece of work, which I personally found very interesting and will undoubtedly be of interest to a wide range of readers.

Reviewer #2 Comments:

The manuscript by Melo et al. reports on a proband with a severe and rare form of heterotopic ossification (HO). The proband had an duplication of ~820 kb from chr2 inserted into chrX. This led to alterations of a TAD and misexpression of a gene on chrX, ARHGAP36. ARHGAP36 is known to inhibit PKA signaling and activate Hedgehog (HH) signaling (although the mechanism(s) involved in the latter are not fully clear with some contradictory data in the literature). RNAseq analyses of control fibroblasts and fibroblasts from the proband, as well as of control and ARHGAP36 transfectants revealed changes to HH and TGFbeta pathway-associated genes. Overexpression of ARHGAP36 in fibroblasts and C2C12 myoblasts reduced TGFbeta signaling. Studies with fibroblasts from the proband and three controls were also studied for responses to TGFbeta, BMP2, and a GLI inhibitor, as well as for in vitro osteogenesis. The authors conclude that misexpression of ARHGAP36 leads to alterations in TGFbeta and HH signaling, which in turn underlie the clinical manifestation of the duplication/insertion.

R: We thank Reviewer #2 for the positive comments about our manuscript, and we do agree that the ARHGAP36 function regarding the hedgehog (HH) signaling is not completely understood. Nonetheless, there is robust evidence in the literature confirming that ARHGAP36 either activates HH directly via suppression of fused (Sufu) or indirectly by inhibiting PKAC (Rack et al., 2014; Eccles et al., 2016). It has been previously shown that ARHGAP36 interacts with the HH pathway by promoting the nuclear translocation of Gli1/2 transcription factors, the main effectors of the canonical HH signaling pathway. This interaction occurs through inhibition of the suppressor of fused (Sufu), which in the absence of a HH ligand binds and represses Gli transcription factors (Zhang et al. 2019; Rack et al. 2014). Moreover, ARHGAP36 is an antagonist of PKA signaling through targeting protein kinase C (PKAC) for ubiquitin dependent proteolysis. Additionally, ARHGAP36 functions as a pseudosubstrate inhibitor of PKA (Eccles et al. 2016). Taken together, there is strong

evidence in the literature supporting the role of ARHGAP36 in HH activation. However we tried to tone down some of our claims in the revised manuscript version to make sure it is clear that a final functional link still needs to be shown.

The genomic analyses leading to the conclusion that ARHGAP36 is the gene critically affected by the duplication/insertion appear to be well performed, but this is not an area of expertise for me, so I defer to reviewers with greater experience. The RNAseq analyses showed changes to pathways that would be expected to be altered in cells with aberrant osteogenesis, and provided useful leads for the signaling studies (along with what is known of ARHGAP36 function). I find, however, that the signaling studies fall short of the level required for firm conclusions.

1. Figure 5: The studies with TGFbeta are not convincing. Although the TGFbeta response in cells overexpressing ARHGAP36, or in proband fibroblasts, trends lower than controls, in most cases it does not rise to the $p < 0.05$ value selected for significance. One major reason for this seems to be the extreme variability of the controls.

R: We agree with the reviewer that some experiments performed showed high relative light units (RLU) variability, therefore we repeated the following experiments with increased number of technical replicates:

- a) TGFβ1 in NIH/3T3 cells: Mock and 0.1 nM, n = 4.
- b) TGFβ1 in C2C12 cells: Mock and 0.1 nM, n =5; 0.2 nM, n =4.

The TGFβ1 pathway analysis by luciferase reporter assay was performed with additional technical replicates to remove the effect of RLU variability observed in some experiments (new Figure 5a). After increasing the number of technical replicates, we observed the same trend as obtained in the experiments performed for the first manuscript submission, but this time the differences were more pronounced: upon TGFβ1 exposure in NIH/3T3 at 0.1 and 0.2 nM TGFβ1 concentrations, we observed a reduced RLU signal in cells transfected with hARHGAP36, this time with significant p value in both concentrations. In C2C12 cells, increasing the number of technical replicates (n = 5) showed a significant reduction ($p < 0.05$) of luciferase activity at 0.1 nM TGFβ1 concentration in both hARGHAP36 and mIgsf1

transfected cells. This information was included in the manuscript page15.

New Figure 5a. Experiments performed with increased technical replicates gave stronger significant results.

This is seen even in panel B, where treatment of C2C12 cells with 0.2 nM BMP2 led to the greatest differential. The 3 controls in this panel differ from each other by 5- to 6-fold in an assay where the signal to noise (+/- ligand) averages 10-fold. These results are not compelling.

R: We agree with the reviewer and we repeated the BMP2 exposure to reduce the high RLU variability. After repeating the experiments, now with lower technical variability, we observed a significant reduction of BMP signaling in both cells (NIH/3T3 and C2C12) at the higher BMP2 concentration (5 nM), in both hARHGAP36 and mlgsf1 transfected cells ($p < 0.01$ and < 0.0001). We changed the text according to the new results on page 15.

Manuscript submission results

Manuscript review results

New Figure S7. This new supplementary figure was replaced by the old Figure S7 in the new manuscript version.

In panel C, using proband-derived fibroblasts is a great approach, and I give the authors credit for using three different controls, but the controls are so different from each other in pSMAD3 levels, it is not possible to conclude anything. Similarly, pSMAD1/5 levels are extremely variable in both proband and controls.

R: Regarding the experiment on Figure 5C, we agree with the reviewer that the controls show high variability from each other in both pSMAD3 and pSMAD1/5 levels. During the revision process, we increased the number of technical replicates (n=6) for measuring pSMAD3 level. We observed a significant reduction (p<0.05) of phosphorylated SMAD3 levels after 60 minutes of TGFβ1 exposure in proband samples compared with controls. We change the text according to the new results on page 15.

New Figure 5c. Increasing the number of technical replicates (n=6) for TGFβ1 induction significantly reduced pSMAD3 phosphorylation levels in proband samples.

Whereas pSMAD1/5 response towards BMP2 is variable, the difference in pSMAD3 upon TGFβ1 stimulation is clear in all technical replicates.

2. Figure S9: This figure has a number of peculiar aspects to it. ARHGAP36 levels drop very quickly after transfection (reason not clear) so the cells were transfected again after 7 days in culture, yet this second large increase in ARHGAP36 levels had essentially no effect on target expression. This goes unmentioned. Additionally, although the fold-difference in expression of COL1A1, etc. is higher at day 1 with ARHGAP36, the overall levels at this time point are rather low, as all these target proteins are induced over the culture period. In fact, looking at the Western blots, it seems all the ARHGAP36 overexpression is doing is very modestly accelerating osteogenesis.

R: We are not sure why GFP protein expression is stable across all time points and ARHGAP36 protein levels drop quickly after transfection. Here we pinpoint few events that could have caused the quick drop of ARHGAP36 levels after transfection: a) the ARHGAP36 ORF contain specific sequences that could be targeted by miRNAs after transfection; b) the ARHGAP36 protein is being target to be degraded by protease complexes, e.g. the ubiquitin–proteasome system (UPS). c) ARHGAP36 high expression in mesenchymal stem cells (MSCs) could cause cell toxicity and it could be inhibited by negative feedback loop. It is not possible from the current perspective to pinpoint what is the most likely cause for ARHGAP36 quick reduction after transfection. We agree with Reviewer #2 that after the second ARHGAP36 transfection the cells did not respond as expected in regards to increasing levels of bone markers such as COL1A1. As the reviewer mentioned: “although the fold-difference in expression of COL1A1, etc. is higher at day 1 with ARHGAP36, the overall levels at this time point are rather low, as all these target proteins are induced over the culture period.”, we do believe that our short-term ARHGAP36 exposure is not sufficient to cumulatively produce extracellular matrix proteins in MSCs, which are well known to be produced over long exposure time during cell culture. This problem could have been tackled by exposing the MSCs with ARHGAP36 expression over weeks, but we did not test this hypothesis in this manuscript. On the other hand, we disagree with the reviewer regarding the impact of ARHGAP36 transfection as causing only a “minor contribution” to

osteogenesis. Based on the experiments on Figure S9 d,e, we observed an increase of 2-3 fold of bone markers upon ARHGAP36 transfection. Further experiments stimulating ARHGAP36 during a longer period could validate or discard this hypothesis.

3. Figure 6: I don't understand panel B. Since the data are reported as fold-change in AR staining, it appears that the control cells did not change at all over 5 weeks (the value is 1 all the way across). In contrast, the proband cells show elevated AR staining at 2-3 weeks (as would be expected), but returns to a value similar to control at 5 weeks. How is this possible, that week 5 is lower than weeks 2-3? Panels D and E are a reasonable beginning to analysis of HH signaling, but use of GANT61 is not a sufficient approach to draw firm conclusions. Also, in panel D, the use of fold-change values obscures whether the proband cells have higher absolute levels of GLI1 expression than control cells – which would be expected if the authors hypotheses are correct.

R: In Figure 6B, we show the AR staining results in a fold-change manner, i.e. we measured the values of each sample per time point, calculated the controls mean values, and compared to the proband ones. After the 4th week of osteogenesis, it is very likely that controls and probands cells reached the osteogenesis saturation of the plate, therefore there's no more difference in AR staining between these samples.

We agree with Reviewer #2 that, although GANT61 is a known GLI blocker and indeed reduced *GLI1* expression in our samples, other chemical compounds such as Vismodegib and Cyclopamine could be better targets to inactivate the HH signaling. RT-qPCR performed in Figure 6D shows a mild increase of *GLI1* (~2.5-fold) in proband cells compared to controls.

Taken together, while I think the identification of the proband's duplication/insertion and the role of ARHGAP36 as the likely key gene are interesting, the mechanistic studies do not provide a compelling view of how this genome perturbation results in HO.

R: While we can be sure that the insertion causes the phenotype in our studied proband and that ARHGAP36 plays a central role in this disease, future experiments using a mouse

model should be performed to dive into the pathomechanism. However, since engineering a mouse model containing a large insertion in a specific locus is technically very challenging and time consuming, and no more patient material is available, we believe it is beyond the scope of this manuscript to decipher the final mechanism of this disease

Reviewer #3 Comments:

The authors have conducted a thorough cytogenetics and genomics approach to discover HO-causing gene in a proband with progressive ectopic ossification. They demonstrated that interchromosomal insertional duplication of 820kb on chromosome 2 insertion at the ARHGAP36/IGSF1 locus in chromosome X resulted in enhancer hijacking thereby increasing expression of ARHGAP36 in proband's fibroblasts. Functional relevance of ARHGAP36 up-regulation in the severe HO case was investigated by overexpression studies using primary cellular systems in vitro. Several candidate signaling pathways identified by RNA-seq and DEG enrichment analysis were examined and, among others, TGF β signaling was found to be marginally inhibited by ARHGAP36 overexpression. Interestingly, despite increased ECM deposition by ARHGAP36-overexpressing cells and accelerated osteogenesis of proband's fibroblasts, hedgehog signaling was not significantly influenced by ARHGAP36 overexpression. Identification of ARHGAP36 as a novel HO-causing gene is intriguing. However, the evidences for disease-causing functionality of ARHGAP36 presented in this study are not convincing enough to directly link it to HO.

R: We are thankful to the Reviewer #3 comments, while we agree with the reviewer that in our current manuscript we did not establish the final functional mechanisms, we do believe that the data presented here supports the hypothesis of ARHGAP36 being involved in this disease: a) it is a large structural variant, inserted in a gene desert region in the chrX, that influences - partially - the X inactivation process; b) such variant activates ARHGAP36; c) RNA-seq from fibroblasts shows dysregulation of important bone-related pathways; d) induced ARHGAP36 expression inhibits both BMP and TGF β pathways; and e) transient ARHGAP36 expression in MSCs activates bone-related markers. In addition, we have to highlight the difficulty of studying HO in vitro which is a multicellular process dependent on unique signaling niches. As for other HO studies, for instance the study of fibroblasts ossificans progressiva (FOP), the real formation of HO is only studied in animal models, whereas cell models only allow to study a simplistic first view on the initial processes which are accumulating to HO. In most of the cases the formation of HO is also dependent on secondary hits (FOP) or additional factors which can be unknown at the current time point.

Main Concerns

(1) Any histology data with proband's biopsy, showing endochondral ossification vs. intramembranous ossification?

R: We have tried several times to obtain biopsies of the affected peri-articular regions, but the parents refused to consent to this invasive procedure. They also did not consent to a

post mortem autopsy. Collecting these unique biological samples would have been great for understanding the molecular mechanism of the disease, however this was not possible.

(2) GTPase activity assay data of proband's ARHGAP36?

R: It has been previously shown that ARHGAP36 and ARHGAP24 are the only RhoGAP-family proteins that show no GTPases response (Muller et al., 2020), therefore we did not measure the GTPase activity in proband's cell.

Figure retrieved from Muller et al., 2020.

(3) In addition to the luciferase assays for TGFβ/BMP signaling, target gene expression of the TGFβ/BMP signaling pathways can support the presented data.

R: We performed RT-qPCR using RNA extracted from proband and control fibroblasts for the following pSMAD2/3 target genes: *PAI1*, *CTGF*, *SNAI1*, *SLUG* and *TWIST*. Whereas *PAI1* and *SNAI1* are equally induced by TGFβ1 in all control and the proband cells, *CTGF* was significantly less expressed in proband samples compared to Control 1 ($p < 0.001$) and Control 3 ($p < 0.05$), thereby following the before described reduced levels of pSMAD3. *SLUG* and *TWIST* were not induced after TGFβ stimulation. Whereas the analysis of TGFβ1 signaling capacity suggests, that the observed reduced TGFβ1 response in proband cells is only a function of natural variation, our CAGA₁₂-Luc data equally highlights that ARHGAP36 overexpression negatively influences TGFβ1-induced pSMAD3 and consequently target gene expression (Figure 5a).

(4) Authors examined canonical TGFβ signaling by looking at phos-SMAD3 levels. I am wondering if non-canonical pathways is influenced by ARHGAP36 overexpression, since TAK1 was known to negatively regulate HH-Gli pathway.

R: During the review process, we performed Western blot analysis to evaluate the non-canonical TGFβ signaling by measuring the phosphorylation levels of TAK1 and p38. We observed no differences in pTAK1 levels of proband cells compared to controls. However, the TAK1-downstream target p38 (pp38) showed significantly elevated levels of p38 phosphorylation in proband cells compared to controls, which was independent of TGFβ1-induced TAK1 phosphorylation. These results were added to the manuscript on pages 15-16 and on Figure S8e.

New Figure S8e.

(5) As authors speculated in Discussion, what if cells are co-stimulated with hedgehog and TGFβ? Would it bring out the phenotype caused by ARHGAP36 misexpression in proband's FB?

R: We agree with Reviewer #3 that co-stimulating HH and TGFβ in wild-type cells would be a great experiment to validate our hypothesis by, in theory, recapitulating the proband's cellular phenotype. However, after stimulating HH in proband and control cells, we did not observe an induction of *GLI1* in control samples (Figure below). The proband sample showed only a mild increase of *GLI1* (~2.5-fold change). Therefore we believe that using the HH molecule to activate the HH signaling in fibroblasts has only minor effects in *GLI1* expression, therefore we believe that this molecule is not the best chemical to study *in vitro* this disease.

(6) PKA downstream substrates were affected (e.g. phosphorylation) by proband's ARHGAP36?

R: We performed Western blot to evaluate PKA downstream substrates but we did not observe variance among proband and control samples based on their phosphorylation levels (Figure below).

(7) Some minor suggestions: In addition to quant data of alizarin red stain for matrix calcification, images of ARS stained cells would be nice.

R: We, unfortunately, did not collect images during the osteogenesis induction.

Reviewer #4 (Remarks to the Author):

The starting point of this topical pathophysiological study is a young patient with an ultra-rare condition characterized by a dramatic and progressive ectopic calcification, leading to early death due to heterotopic ossification, which is aberrant ossification of connective tissue.

Using extensive mechanistic studies, it has been shown that this condition is caused by an inter-chromosomal insertional duplication that disrupts a topologically associating domain of the X chromosome. This results in 'enhancer hijacking' and ARHGAP36 misexpression. ARHGAP36 activation has been shown to have an impact on gene expression of key pathways implicated in bone formation and heterotopic ossification.

The conclusions of the study are supported by extensive and well-conducted experimental work including functional studies on biologically relevant patient material, which is a strength. A potential limitation of the study is that it deals with one single case having an ultra-rare disease. It would be interesting to know if attempts have been done to find additional cases f.i. using GeneMatcher. This minor weakness is balanced by the excellent functional work supporting the proposed disease mechanism.

Overall, this is a unique study demonstrating that disruption of the 3D genome and enhancer hijacking caused by an inter-chromosomal duplication activates ARHGAP36, leading to soft tissue to bone transformation.

R: We thank Reviewer #4 for the positive comments towards our manuscript. We did search for additional patients with ARHGAP36 mutations or progressive ossification phenotype within our collaboration network, with experts in the field of pediatric bone malformations, as well as peers in International meetings (e.g., ESHG, ASHG). Despite our continuous attempts, in almost ten years no similar case was reported, in line with the peculiar gain of function mechanism described in our manuscript. We also submitted ARGAPH36 to GeneMatcher and obtained 11 "matches". However, none of the records clinically fits with the features presented in our proband, where most of the reported patients showed CNS involvement, DD/ID, behavior, dystonia, among other symptoms.

There are some minor points to address:

1/Abstract: The abbreviation TAD could already be introduced in the abstract

R: Done.

2/ Introduction: the references jump from 12 (Introduction) to 59 (Methods)

R: Corrected.

Methods

3/ IRB number of the research protocol?

R: This project was approved by the internal review board committee of the University of Torino, Dep. Medical Sciences under the protocol number 0053916. This information was added to the manuscript on page

4/ Cell culture and samples: is RNA total RNA?

R: Total RNA. Corrected in the manuscript.

5/ p.6: how has the specificity of the antibodies been assessed?

R: The antibodies used in this work were used many times in previous projects and publications from our laboratory, therefore they were all validated on previous works. Most of the antibodies were bought from Cell Signaling and are widely accepted in the field as indicated by high amount of citing articles (e.g. Phospho-Smad3 (Ser423/425) (C25A9) Rabbit mAb #9520 Citations (690); Phospho-p38 MAPK (Thr180/Tyr182) (28B10) Mouse mAb #9216, Citations (491)).

6/ array CGH, array-CGH: only abbreviation mentioned

R: We included "Array-based comparative genomic hybridization" and kept "array-CGH" throughout the manuscript.

7/ in situ hybridization: in situ in italics

R: Done.

8/ p.6: the title 'Cytogenetics and genomics' does not cover Hi-C and RT-qPCR

R: We included "transcriptomics" in the title.

9/ RNA-seq and RNA-Seq

R: We thank the reviewer for checking this typo and we kept "RNA-seq" throughout the manuscript.

10/ p.7: Exome and Genome sequencing: already abbreviated

R: Corrected.

11/ Hochmann et al. 2022: status of the manuscript?

R: The Hochmann et al. 2022 manuscript is still under revision in Science Translational Medicine, which might take longer to be published.

12/ In vitro osteogenic differentiation in proband cells is used as an interesting model for bone formation in a dish. Could be explained why induced fibroblasts are used, and why no reprogramming (iPSC) and differentiation is needed?

R: Fibroblasts and mesenchymal stem cells (MSCs) are the most used *in vitro* source to study heterotopic ossification. In fact, fibroblasts share many characteristics with MSCs, for instance, they are indistinguishable in their morphology, share the same immunophenotype and have the ability to undergo adipogenesis, osteogenesis and chondrogenesis (Brohem et al. 2013). Moreover, it has been shown that fibroblasts can obtain osteogenic and chondrogenic mesenchymal fates in extraskeletal locations during trauma induced heterotopic ossification (Plikus et al. 2021; Cappato et al. 2020). Due to the fact we had access to the proband's fibroblasts, an ideal cell type to study this disease, we did not generate iPSC to further differentiate into other cell types.

Results

13/ Abbreviations: osteogenic media (OM)(p.10): OM is already used before; heterotopic ossification is abbreviated (HO) but the abbreviation is not used on p.10 (title) and p.11 (text). MRI: already abbreviated before?; p.10: abbreviation MSC used but only explained on p.12.

R: We thank the reviewer for carefully checking abbreviations and we corrected all mentioned mistakes in the manuscript.

14/ functionally relevant genes are based on the clinical presentation: how is this defined?

R: We evaluated genes-members of the BMP-TGF β , WNT and HH signaling, known pathways related to bone-related diseases.

15/ p.11: open reading frame instead of 'open read frame'

R: Corrected.

16/ p.12: The chr2 duplicated region partially maintainS

R: Corrected.

17/ p.13: we excluded skewED X chromosome inactivation.

R: Corrected.

18/ p.13: gene body: is the transcriptional unit meant?

R: We meant the gene sequence, containing exons, introns, 3' UTR, but lacking the open reading frame, for the reason that the promoter and the first *ANTRX1* exons are not included in the duplication.

19/ CRUP: abbreviated on p.8 and p.13

R: Corrected.

20/ p.13: differentiation via HH pathway: via 'the' HH pathway

R: Corrected.

21/ p.15: We observed *a* slightly reduced (...) levels: remove 'a'

R: Corrected.

22/ p.15: (...) although *we* did not reach significance: 'we' is missing

R: Corrected.

23/ p.15: Title: shows increased ECM deposition

R: Corrected.

Discussion

24/ p.16: by 'an' enhancer hijacking mechanism

R: Corrected.

25/ p.19: death associated protein kinase 1 (DAPK1)

R: Corrected.

Acknowledgements

26/ the name Beatrice is mentioned, assuming the parents consented with this.

R: We removed the name Beatrice from the acknowledgements.

Figures

27/ Figure 1: both Array-CGH and Array CGH used.

R: Corrected.

Supplementary Information

28/ Table S1 (several genes associated to heterotopic calcification/ossification disorders): mentioned on p.11 but could not be found in the Supplementary Information.

R: We apologize for this mistake. We noticed that Table S1 was not included in the Supplementary Information during the first submission. We corrected it and included Table S1 in the resubmission.

29/ Duplicated genes excluded as candidates to explain the phenotype. Although it is extensively motivated why several duplicated genes can be excluded as candidate genes for

the phenotype, it should be mentioned that this is speculative. References are missing for some genes. Decipher with capital. *Drosophila*: to capitalize and italicize.

R: We agree with the reviewer and we stated in Supplementary Information that excluded the duplicated genes as candidates were speculative. We corrected reviewer requests.

30/ p.3: skewED X inactivation; chromosome 2 instead of chromosomeS; showed AN increased number of maternal reads; de novo: italicize.

R: Corrected.

31/ p.5: Two regimes ARE observed

R: Corrected.

References

Brohem, CA, CM De Carvalho, CL Radoski, FC Santi, MC Baptista, BB Swinka, C de A. Urban, LRR De Araujo, RM Graf, and IHS Feferman. 2013. "Comparison between fibroblasts and mesenchymal stem cells derived from dermal and adipose tissue." *International journal of cosmetic science* 35 (5): 448-457.

Cappato, Serena, Riccardo Gamberale, Renata Bocciardi, and Silvia Brunelli. 2020. Genetic and acquired heterotopic ossification: a translational tale of mice and men. *Biomedicines* 8 (12): 611.

Chaisson MJP, Sanders AD, Zhao X, Malhotra A, Porubsky D, Rausch T, Gardner EJ, Rodriguez OL, Guo L, Collins RL et al. 2019. Multi-platform discovery of haplotype-resolved structural variation in human genomes. *Nat Commun* 10: 1784.

Eccles, R.L., et al., Bimodal antagonism of PKA signalling by ARHGAP36. *Nature communications*, 2016. 7(1): p. 1-16.

Rack, P.G., et al., Arhgap36-dependent activation of Gli transcription factors. *Proceedings of the National Academy of Sciences*, 2014. 111(30): p. 11061-11066.

Ebert P, Audano PA, Zhu Q, Rodriguez-Martin B, Porubsky D, Bonder MJ, Sulovari A, Ebler J, Zhou W, Serra Mari R et al. 2021. Haplotype-resolved diverse human genomes and integrated analysis of structural variation. *Science* 372.

Melo US, Schöpflin R, Acuna-Hidalgo R, Mensah MA, Fischer-Zirnsak B, Holtgrewe M, Klever MK, Türkmen S, Heinrich V, Pluym ID, Matoso E, Bernardo de Sousa S, Louro P, Hülsemann W, Cohen M, Dufke A, Latos-Bieleńska A, Vingron M, Kalscheuer V, Quintero-Rivera F, Spielmann M, Mundlos S. Hi-C Identifies Complex Genomic Rearrangements and TAD-Shuffling in Developmental Diseases. *Am J Hum Genet*. 2020 Jun 4;106(6):872-884. doi: 10.1016/j.ajhg.2020.04.016. Epub 2020 May 28. PMID: 32470376; PMCID: PMC7273525.

Müller, P.M., Rademacher, J., Bagshaw, R.D. et al. Systems analysis of RhoGEF and RhoGAP regulatory proteins reveals spatially organized RAC1 signalling from integrin adhesions. *Nat Cell Biol* 22, 498–511 (2020).

Plikus, Maksim V, Xiaojie Wang, Sarthak Sinha, Elvira Forte, Sean M Thompson, Erica L Herzog, Ryan R Driskell, Nadia Rosenthal, Jeff Biernaskie, and Valerie Horsley. 2021. "Fibroblasts: Origins, definitions, and functions in health and disease." *Cell* 184 (15): 3852-3872.

Zhang, B., et al., Patched1–ArhGAP36–PKA–Inversin axis determines the ciliary translocation of Smoothed for Sonic Hedgehog pathway activation. *Proceedings of the National Academy of Sciences*, 2019. 116(3): p. 874-879.

REVIEWER COMMENTS

Reviewer #1 (Remarks to the Author):

In general, I was very supportive of this manuscript and my previous comments essentially asked for some further analysis to firm up the enhancer highjacking hypothesis and to soften to "firmness" of the conclusions.

The authors have addressed my comment and I have no further comments.

Reviewer #2 (Remarks to the Author):

The authors have responded to my comments with a combination of additional experimentation and explanation.

1. Previous point 1 (Figure 5): The authors have successfully addressed this comment with new Figures 5a, 5c, and S7.

2. Previous point 2 (Figure S9): The authors have provided additional explanation here but the figure is still presented in an awkward fashion. I am not asking for an explanation of why ARHGAP36 levels drop quickly after transfection (it is presumably something related to ectopic expression and possibly irrelevant to its normal regulation). I pointed out that the cells are unresponsive to the second wave of transfection/overexpression and that this was not mentioned in the text. I'm not even sure why they stuck with the 2X transfection under these circumstances. What this leads to is a situation where the cells appear to respond only to the first short wave of ARHGAP36 overexpression. This is the time points the authors refer to in the rebuttal, saying that there is a 2-3 – fold increase of bone markers. None of this is commented on the paper, making the figure complex and ultimately hard to understand (I note that the Figure S9 legend mentions the re-transfection but the fact that it had no effect on marker expression is not brought up). I also note that the GAPDH levels that are used as a loading and expression control clearly rise though the time course. The figure is important for the story. It would be my recommendation that the experiments be repeated with a single ARHGAP36 transfection and a better loading control, with the conclusion that as long as there is high level ARHGAP36 expression, the cultures make higher levels of bone markers, but when it tapers off, the control cultures catch up. Minimally, the text needs to reflect what is in the figures, but that part of the text is still suboptimal for readers.

3. Previous point 3 (Figure 6): If I understand the authors' rebuttal correctly, the control values at each time point in Figure 6b were set to 1.0, and the proband values were plotted as fold-change relative to the controls at the corresponding time point. If that is correct, it explains why the control cultures are always at 1.0 when they should be rising through the time course. It can also explain why the proband cultures' values return to 1.0, as they are simply saturated earlier than the controls. I find this a very awkward way of presenting these data, and it is likely to confuse readers. I suggest that the authors set the control value at week 0 to 1.0, and all other values be plotted as fold-change relative to that single value. It would reflect the overall changes in all the cultures and be a more intuitive way to plot the data.

Reviewer #3 (Remarks to the Author):

The authors made quite an effort to address the issues/comments raised by myself and other reviewers in a reasonably possible ways. Although there still are gaps to fill (especially involved signaling pathways and clear mechanistic details), it may be possible via further study in the future, as the authors mentioned in the letter.

One minor comment: They saw the significant activation of p38MAPK in the proband's cells in the follow-up experiment addressing my comment on non-canonical TGFB signaling via TAK1. They did not comment about potential implication of the results in discussion (e.g. other signaling via Rho-p38MAPK axis).

Reviewer #4 (Remarks to the Author):

The referees' comments have been addressed very well. The rebuttal is convincing, the revision is thorough with new experimental evidence (4C, Western blot, additional in vitro assays).

Reviewer #1 Comments:

In general, I was very supportive of this manuscript and my previous comments essentially asked for some further analysis to firm up the enhancer highjacking hypothesis and to soften to "firmness" of the conclusions.

The authors have addressed my comment and I have no further comments.

R: We thank the Reviewer #1 for the positive comments regarding our manuscript.

Reviewer #2 Comments:

The authors have responded to my comments with a combination of additional experimentation and explanation.

1. Previous point 1 (Figure 5): The authors have successfully addressed this comment with new Figures 5a, 5c, and S7.
2. Previous point 2 (Figure S9): The authors have provided additional explanation here but the figure is still presented in an awkward fashion. I am not asking for an explanation of why ARHGAP36 levels drop quickly after transfection (it is presumably something related to ectopic expression and possibly irrelevant to its normal regulation). I pointed out that the cells are unresponsive to the second wave of transfection/overexpression and that this was not mentioned in the text. I'm not even sure why they stuck with the 2X transfection under these circumstances. What this leads to is a situation where the cells appear to respond only to the first short wave of ARHGAP36 overexpression. This is the time points the authors refer to in the rebuttal, saying that there is a 2-3 – fold increase of bone markers. None of this is commented on the paper, making the figure complex and ultimately hard to understand (I note that the Figure S9 legend mentions the re-transfection but the fact that it had no effect on marker expression is not brought up). I also note that the GAPDH levels that are used as a loading and expression control clearly rise though the time course. The figure is important for the story. It would be my recommendation that the experiments be repeated with a single ARHGAP36 transfection and a better loading control, with the conclusion that as long as there is high level ARHGAP36 expression, the cultures make higher levels of bone markers, but when it tapers off, the control cultures catch up. Minimally, the text needs to reflect what is in the figures, but that part of the text is still suboptimal for readers.

R: We thank the Reviewer #2 for her/his comments regarding Figure S9. In order to make this figure uncomplicated and easily readable, we decided to remove the re-transfection data, as it did not influence the results and interpretation, plus the cells did not respond the same way after the second transfection as they did in the first one. Therefore we generated a new Figure S9 (please find it below).

3. Previous point 3 (Figure 6): If I understand the authors' rebuttal correctly, the control values at each time point in Figure 6b were set to 1.0, and the proband values were plotted as fold-change relative to the controls at the corresponding time point. If that is correct, it explains why the control cultures are always at 1.0 when they should be rising through the time course. It can also explain why the proband cultures' values return to 1.0, as they are simply saturated earlier than the controls.

R: We thank Reviewer #2 for her/his comment regarding Figure 6b. The understanding from Reviewer #2 is correct, the control values were set to 1.0 at each time point. We included the following phrase in page 16: "The control values at each time point were set to 1.0, and the proband values were plotted as fold-change relative to the controls at the corresponding weekly time point." We do believe that with this clarification, the Figure 6b will be easily interpreted by readers.

I find this a very awkward way of presenting these data, and it is likely to confuse readers. I suggest that the authors set the control value at week 0 to 1.0, and all other values be plotted as fold-change relative to that single value. It would reflect the overall changes in all the cultures and be a more intuitive way to plot the data.

R: We understand the point raised by Reviewer #2 regarding Figure 6b, however, the visualization graph we selected to show the results is, in our opinion, the best option based on the collected data. We could show this data in the format suggested by the reviewer, however, as you can see in the figure below, we had a huge discrepancy in measuring the

alizarin red (AR) plates on week 2: the overall AR staining values at week 2 were extremely low, which makes the requested visualization look strange.

Figure 6b. Left panel original figure. Right panel, suggestion from the Reviewer #2.

Because the measured AR staining during week 2 shows a different overall values pattern from the other time points, we do believe that showing the graph in a fold-change way would be the best way to show the data.

Reviewer #3 Comments:

The authors made quite an effort to address the issues/comments raised by myself and other reviewers in a reasonably possible ways. Although there still are gaps to fill (especially involved signaling pathways and clear mechanistic details), it may be possible via further study in the future, as the authors mentioned in the letter.

R: We are thankful to the Reviewer #3 comments.

One minor comment: They saw the significant activation of p38MAPK in the proband's cells in the follow-up experiment addressing my comment on non-canonical TGFB signaling via TAK1. They did not comment about potential implication of the results in discussion (e.g. other signaling via Rho-p38MAPK axis).

R: We included the following phrase in the discussion part (page 20) regarding the potential implication of other signaling pathways: "Further, we observed independent of TGFβ1-stimulation an enhanced baseline phosphorylation of p38 in proband cells. As p38 plays a pivotal role in different steps of osteoblast differentiation, mainly through induction of pro-osteogenic transcription factors like RUNX2, it remains an interesting downstream effector of ARHGAP36-dependent enhanced pro-osteogenic capacities. While p38 activation can be induced by a variety of upstream triggers including different biochemical and mechanical signaling cascades and cellular stresses, future studies should evaluate the link between ARHGAP36 function and p38 activation."

Reviewer #4 (Remarks to the Author):

The referees' comments have been addressed very well. The rebuttal is convincing, the revision is thorough with new experimental evidence (4C, Western blot, additional in vitro assays).

R: We thank the Reviewer #4 for the positive comments regarding our manuscript.

REVIEWERS' COMMENTS

Reviewer #2 (Remarks to the Author):

The authors have addressed my final comments in a satisfactory manner, I congratulate them on an interesting study.

Reviewer #2 (Remarks to the Author):

The authors have addressed my final comments in a satisfactory manner, I congratulate them on an interesting study.

R: We are very thankful to Reviewer #2 for all her/his comments and suggestions that substantially improved our manuscript.